# Xyloglucan processing machinery in *Xanthomonas* pathogens and its role in the transcriptional activation of virulence factors

Plinio S. Vieira [1,9], Isabela M. Bonfim [1,2,9], Evandro A. Araujo[1,3], Ricardo R. Melo[1], Augusto R. Lima[1], Melissa R. Fessel[4], Douglas A. A. Paixão[1], Gabriela F. Persinoti [1], Silvana A. Rocco[5], Tatiani B. Lima[1], Renan A. S. Pirolla[1], Mariana A. B. Morais [1], Jessica B. L. Correa[1], Leticia M. Zanphorlin[1], Jose A. Diogo[1,2], Evandro A. Lima[1], Adriana Grandis[6], Marcos S. Buckeridge[6], Fabio C. Gozzo[7], Celso E. Benedetti[5], Igor Polikarpov [8], Priscila O. Giuseppe [1✉] & Mario T. Murakami [1✉]

Xyloglucans are highly substituted and recalcitrant polysaccharides found in the primary cell walls of vascular plants, acting as a barrier against pathogens. Here, we reveal that the diverse and economically relevant *Xanthomonas* bacteria are endowed with a xyloglucan depolymerization machinery that is linked to pathogenesis. Using the citrus canker pathogen as a model organism, we show that this system encompasses distinctive glycoside hydrolases, a modular xyloglucan acetylesterase and specific membrane transporters, demonstrating that plant-associated bacteria employ distinct molecular strategies from commensal gut bacteria to cope with xyloglucans. Notably, the sugars released by this system elicit the expression of several key virulence factors, including the type III secretion system, a membrane-embedded apparatus to deliver effector proteins into the host cells. Together, these findings shed light on the molecular mechanisms underpinning the intricate enzymatic machinery of *Xanthomonas* to depolymerize xyloglucans and uncover a role for this system in signaling pathways driving pathogenesis.

---

[1] Brazilian Biorenewables National Laboratory (LNBR), Brazilian Center for Research in Energy and Materials (CNPEM), Campinas, São Paulo, Brazil. [2] Graduate Program in Functional and Molecular Biology, Institute of Biology, University of Campinas, Campinas, São Paulo, Brazil. [3] Brazilian Synchrotron Light Laboratory (LNLS), Brazilian Center for Research in Energy and Materials (CNPEM), Campinas, São Paulo, Brazil. [4] Butantan Institute, Butantan Foundation, São Paulo, São Paulo, Brazil. [5] Brazilian Biosciences National Laboratory (LNBio), Brazilian Center for Research in Energy and Materials (CNPEM), Campinas, São Paulo, Brazil. [6] Department of Botany, Institute of Biosciences, University of São Paulo, São Paulo, Brazil. [7] Institute of Chemistry, University of Campinas, Campinas, São Paulo, Brazil. [8] São Carlos Institute of Physics, University of São Paulo, São Carlos, São Paulo, Brazil. [9] These authors contributed equally: Plinio S. Vieira, Isabela M. Bonfim. ✉email: priscila.giuseppe@lnbr.cnpem.br; mario.murakami@lnbr.cnpem.br

Xyloglucans (XyGs) comprise a class of highly complex polysaccharides present in the primary cell wall of vascular plants from clubmosses to angiosperms, including all agricultural cultivars[1]. These recalcitrant polysaccharides form an intricate network with cellulose, which is critical for cell wall function and structure, and serves as a physical barrier against pathogen invasion and colonization[2].

XyGs are structurally and chemically diverse, consisting of a β-1,4-linked glucan backbone decorated with α-1,6-xylosyl residues, which might have additional decorations such as D-galactose, L-fucose, and L-arabinose, depending on the source at the tissue level in plants[3]. These polysaccharides can also be acetylated and this modification is known to affect their physicochemical properties and interaction with other cell-wall components[1,3,4].

To cope with XyGs, many microorganisms, such as saprophytes[5] and commensal bacteria from the human gut[6], harbor enzymatic toolboxes encoded by a set of physically linked genes known as XyG utilization loci (XyGUL). Plant pathogens from the *Xanthomonas* genus also encompass a gene cluster predicted to degrade XyGs (Fig. 1). These pathogens exhibit a high tissue and host specificity, colonizing mesophylls or xylem vessels of over 400 distinct monocotyledons and dicotyledons, including many economically important plants such as citrus, cotton, and corn[7,8]. Regardless of the lifestyle and ecological niche specialization, most *Xanthomonas* species harbors this predicted XyGUL in their genomes, indicating the relevance of this system for these bacteria. However, the molecular mechanisms underpinning XyG depolymerization and potential biological roles in *Xanthomonas* and other phytopathogens remain so far elusive.

Therefore, here we used the causal agent of citrus canker *Xanthomonas citri* pv. *citri* (*X. citri*) as a model organism[9] to investigate the molecular basis of XyG breakdown and its potential involvement in pathogenesis and host-pathogen interactions. Our results show that *Xanthomonas* XyGUL encodes a highly elaborate enzymatic cascade including distinct activities (acetylesterase, α-L-fucosidase, β-galactosidase, α-xylosidase, and xyloglucanase), catalytic mechanisms (inverting and retaining), modes of action (endo and exo), and 3D architectures (multimodular and quaternary arrangements). This machinery notably differs from other known XyGULs, expanding the current knowledge about microbial molecular strategies associated with the depolymerization and utilization of recalcitrant plant polysaccharides. Furthermore, we reveal a link between this enzymatic system and bacterial virulence through a stimulatory effect of its products on the expression of several key virulence factors, including the type III secretion system (T3SS), a needle-like apparatus that inject effector proteins into the plant cell to modulate host responses in favor of bacterial colonization[10].

## Results

### The XyGUL gene architecture in *Xanthomonas* diverges from Bacteroidetes. Genomic analysis revealed that most *Xanthomonas* species conserve a predicted XyGUL consisting of two outer membrane TonB-dependent transporters (TBDTs), four glycoside hydrolases belonging to the families GH74, GH31, GH35, GH95 and one esterase with no significant similarity with any carbohydrate esterase family (Fig. 1a, and Supplementary Tables 1 and 2). Adjacent to this cluster, there are common components for xylose metabolism (D-xylulokinase and xylose isomerase) and an inner membrane MFS sugar transporter (Supplementary Figs. 1 and 2). A search in the Polysaccharide Utilization Loci database[11] did not result in any similar organization in Bacteroidetes, except for the clustering of three or two XyG-related genes in some species. In characterized *Bacteroides*

XyGULs, the clustering of GH31 and GH95 genes has been observed, but in association with other carbohydrate-active enzymes (CAZymes)[6,12]. The synteny of GH31, GH35, and GH95 genes was reported in the XyGUL from the saprophyte *Celvibrio japonicus*[5], but not physically linked to endoxyloglucanases or esterases genes.

The XyGUL is highly conserved across the *Xanthomonas* genus regardless of the broad range of hosts (monocotyledons and dicotyledons) and tissue specificity (mesophyll or xylem vessels) (Fig. 1b). Few of them have lost the predicted GH74 xyloglucanase, but endoglucanases encoded outside the XyGUL, such as GH5, GH9, and GH12 members, may compensate for its absence (Supplementary Fig. 2). The only exceptions that lack most of XyGUL genes are *Xanthomonas* species colonizing gramineous monocotyledons such as *X. oryzae* (rice), *X. translucens* (wheat), and *X. albilineans* (sugarcane) (Fig. 1b). This apparent loss of XyG-degrading capacity correlates with the typically lower contents of XyG in the cell walls of these plants[13–16]. Furthermore, many *Xanthomonas* species carrying the predicted XyGUL are promoters of several diseases in highly relevant agricultural crops such as corn (*X. vasicola* pv. *vasculorum* causing bacterial leaf streak), tomato (*X. perforans* causing bacterial spot), banana (*X. campestris* pv. *musacearum* causing enset wilt), citrus (*X. citri* pv. *citri* causing citrus canker), and cabbage (*X. campestris* pv. *campestris* causing black rot)[17] (Fig. 1b, and Supplementary Table 3). These observations indicate that this system might play important roles in supporting a successful infection, which led us to investigate in depth the molecular mechanisms governing XyG processing by *Xanthomonas* and its potential biological functions.

### XyGUL endo-enzyme exploits arginine–carbohydrate interactions. The first enzymatic unit of the *Xanthomonas* XyGUL is encoded by XAC1770 (named here as *Xac*Xeg74) and belongs to the GH74 family, which is known to have high specificity for XyGs[18,19]. According to kinetic characterization and cleavage pattern analysis, *Xac*Xeg74 is an endo-dissociative enzyme generating a broad distribution of xyloglucan oligosaccharides (XyGOs), but preferentially Glc4- and Glc3-based products (Fig. 2a, and Supplementary Fig. 3c). These results indicate that *Xac*Xeg74 would accept both X (α-D-Xyl*p*-(1,6)-β-D-Glc*p*-(1-), and G (-4)-β-D-Glc*p*-(1-) motifs at the -1 subsite, correlating with the presence of a glycine residue (G465) that confers such capacity to group I members of the GH74 family[19] (Fig. 2b).

To get further insights into XyG recognition by the *Xanthomonas* GH74 enzyme, the crystal structure of *X. campestris* pv. *campestris* GH74 enzyme (*Xcc*Xeg74, sharing 84% of sequence identity with *Xac*Xeg74) was determined in complex with the disaccharide XG spanning the subsites +1 and +2 (Supplementary Fig. 3a, b, and Supplementary Tables 4 and 5). Structural comparisons with the closest structurally characterized xyloglucanases from *Caldicellulosiruptor lactoaceticus* (*Cl*GH74A[19], PDB ID 6P2M and SeqID 39.8%) and *Niastella koreensis* (*Nk*GH74[19], PDB ID 6P2L, and SeqID 34.2%) revealed remarkable differences in the molecular basis for substrate recognition. *Cl*GH74A and *Nk*GH74 rely on several CH–π interactions for substrate anchoring, including at least four aromatic residues at the subsites −3, +1, +3 and +5, and possibly two additional aromatic residues forming the subsites −4 (W126) and −5 (W599) in *Cl*GH74A (Fig. 2b). However, except for the −3 (Y128) and the +1 (W396) aromatic-based subsites, the other aromatic residues are absent in *Xanthomonas* GH74 enzymes, in particular at the +3 and +5 subsites known to be critical for the endo-processive mode of action[19] (Fig. 2b). The lack of aromatic residues in the subsites +3 and +5 of *Xcc*Xeg74

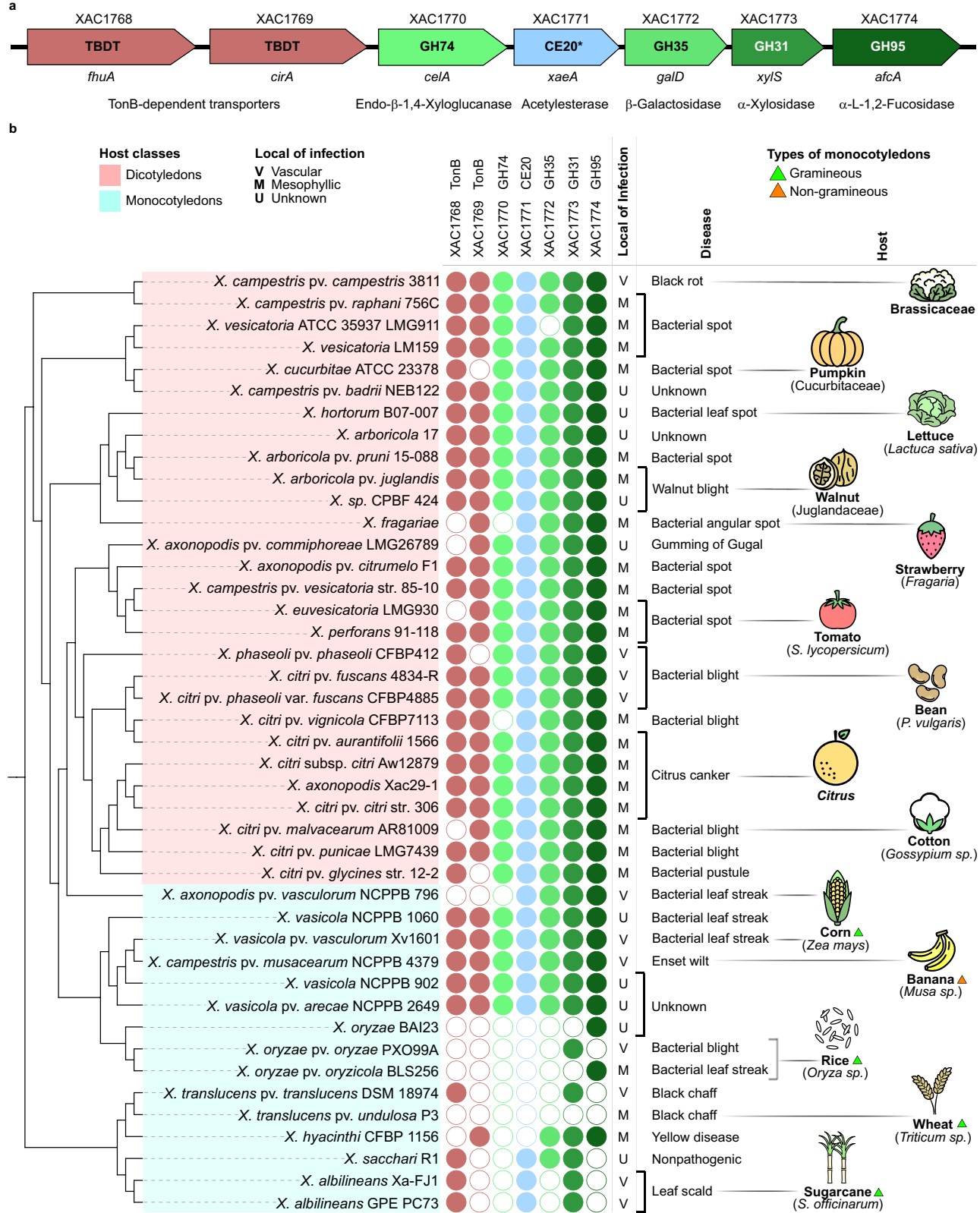

**Fig. 1 The XyGUL conservation in *Xanthomonas* spp. a** XyGUL predicted in *X. citri* pv. *citri* 306 genome showing TonB-dependent transporters (TBDT in pink), glycoside hydrolases (GHs in shades of green) and a carbohydrate esterase (CE in blue, * revealed in this work). **b** Dendrogram of *Xanthomonas* species based on phylogenetic analysis (details in Supplementary Fig. 2) showing the presence (filled circles) or absence (open circles) of XyGUL genes and information about disease and tissue/host specificity. V = vascular, M = mesophyllic and U = unknown. Some species infect dicotyledons (pink box), whereas others colonize monocotyledons (blue box). Monocotyledons are subdivided in gramineous (green triangle) or non-gramineous (orange triangle).

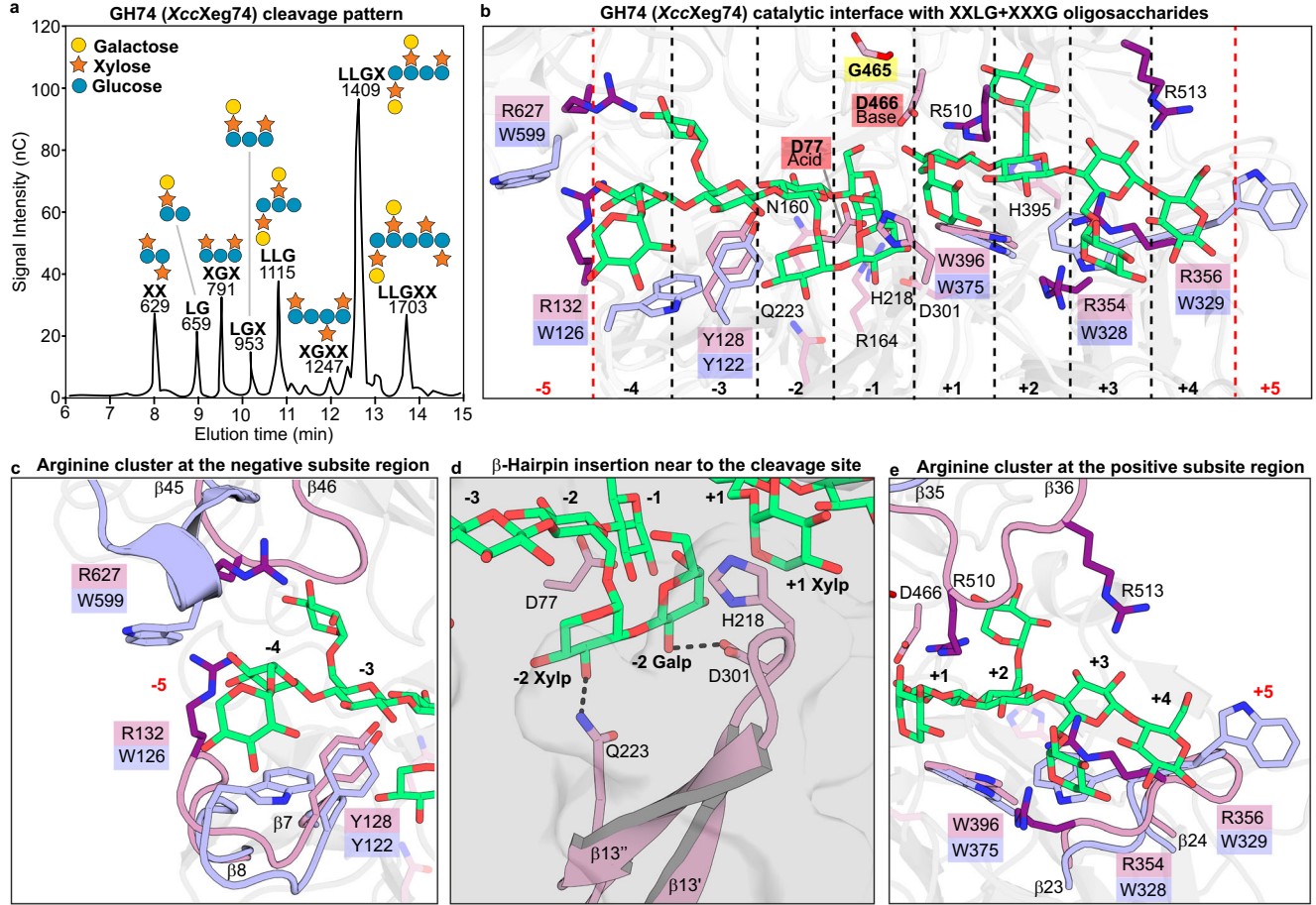

**Fig. 2 The structural determinants for substrate recognition in *Xanthomonas* GH74 endo-β-1,4-xyloglucanases. a** HPAEC-PAD analysis of products released from *Copaifera langsdorfiii* XyG by *Xcc*Xeg74. Numbers above the peaks represent *m/z* values of each product assessed by mass spectrometry (details in Supplementary Fig. 3c). Letters indicate the type of substitutions appended to the glucose backbone: G = non-substituted glucose, X = glucose substituted with a xylose at C-6, and L = X with a galactose appended at xylose C-2. Symbols represent glucose (blue circle), xylose (orange star), and galactose (yellow circle). **b** *Xcc*Xeg74 crystallographic structure (carbon atoms in dark and light purple) superimposed onto the *C. lactoaceticus* GH74 enzyme (carbon atoms in light blue; PDB ID 6P2M[19]), and *N. koreensis* GH74 in complex with XyGOs (only XyGOs shown; carbon atoms in green; PDB ID 6P2L[19]) highlighting the substrate-binding subsites (−4 to +4 in *Xcc*Xeg74 and −5 to +5 in *Cl*GH74A). Residues involved in substrate recognition are shown as sticks, with the labels of catalytic residues from *Xcc*Xeg74 highlighted with red boxes and that of the glycine featuring acceptance to X and G motifs indicated with a yellow box. Note the arginine residues (carbon atoms in dark purple) populating the several substrate-binding subsites in *Xcc*Xeg74. **c**, **d**, **e** Amplified view of specific zones of the GH74 catalytic interface highlighting structural adaptations found in *Xcc*Xeg74 that are divergent from *Cl*GH74A.

is, therefore, consistent with its endo-dissociative mode of action on XyG.

*Xcc*Xeg74 contains three conserved motifs that could compensate for the lack of aromatic platforms in specific subsites: two arginine residues (R132 and R627) at the −4 subsite, a β-hairpin (residues A211-G226) inserted in the η2-β14 loop at the N-terminal lobe, and an arginine cluster at the positive subsite region (Fig. 2b–e, and Supplementary Fig. 4). The β-hairpin interacts concurrently with both galactosyl and xylosyl decorations at the −2 position and with the +1 xylosyl moiety via stacking contacts with H218 (Fig. 2d). In addition, the arginine residue (R510) stacks with the XyG backbone at the +2 position, and the other three nearby conserved arginine residues (R354, R356, and R513) are strategically located to establish polar and/or stacking interactions with the saccharide at the +3 subsite (Fig. 2e). This molecular strategy of carbohydrate recognition based on arginine residues observed in *Xanthomonas* GH74 enzymes is a distinguishing feature among GH families[20], which typically rely on aromatic CH–π interactions for carbohydrate binding.

**The uptake of XyGOs is mediated by TonB-dependent transporters.** After the extracellular cleavage of XyG backbone, the following reactions for the deacetylation and breakdown of the released oligosaccharides likely occur at the periplasm, either by the action of free or membrane-anchored enzymes, as indicated by signal peptide analysis and subcellular localization predictions (Supplementary Table 1). Supporting this hypothesis, the knockout of both TonB-dependent transporters (TBDTs) encoded by the XyGUL (XAC1768 and XAC1769) was highly detrimental to *X. citri* growth with XyGOs as carbon source, but not to the growth with a mixture of its monosaccharides, indicating that depolymerization of XyGOs occurs after passing the outer-membrane using specific TBDTs. Individual knockout of these transporters supports a major role for XAC1769 in XyGOs uptake. In the absence of XAC1768, XAC1769 was sufficient to maintain the bacterial growth akin to the wild-type strain using XyGOs as carbon source, but the opposite was not observed. The knockout of XAC1769 impaired the growth in the late log phase, indicating the importance of this transporter as the XyGOs concentration decreases in the culture medium (Supplementary Fig. 5).

**XyGOs deacetylation by *Xanthomonas* involves a distinctive acetylesterase.** *Xanthomonas* XyGUL harbors a putative esterase gene (XAC1771) initially annotated as sialate 9-*O*-acetylesterase due to the low similarity with current known carbohydrate esterase families (Supplementary Table 2). The presence of an esterase is in accordance with the fact that fucosylated XyGs from dicotyledons can be acetylated at O6 position of the galactosyl moiety[21]. Acetylation modifies the physical and chemical properties of carbohydrates, limiting enzyme accessibility. Therefore, acetate removal is a key step toward efficient processing of this polysaccharide by downstream glycoside hydrolases.

The *Xanthomonas* XyGUL esterase, named here *Xac*XaeA (xyloglucan acetylesterase), showed high activity on pNP-acetate and did not accept moieties longer than acetyl as substrates (Supplementary Tables 6–8, and Supplementary Fig. 6h). It is specific for *O*-acetylation since it was not capable of cleaving *N*-acetylated carbohydrates (Supplementary Fig. 7). However, *Xac*XaeA showed activity on a broad range of *O*-acetylated mono- and disaccharides and did not show a positional preference for acetylated oxygens (Supplementary Fig. 7). As expected, *Xac*XaeA was active towards cell wall extracted xyloglucan oligosaccharides, deacetylating distinct types of structures such as XXLG/XLXG, XXFG, and XLFG (Supplementary Figs. 8 and 9, and Supplementary Table 9).

To get insights into the modular structure and molecular determinants for *O*-acetyl esterase activity, the crystallographic structure of *Xac*XaeA was solved using $Zn^{2+}$ single-wavelength anomalous dispersion (SAD) (Supplementary Table 4). The catalytic domain displays the SGNH hydrolase fold (Fig. 3a–c) found in several lipases as well as esterases[22] from families CE2[23], CE3[24], CE6[25], CE12[26], and CE17[27] (Fig. 3d). However, it is composed of two halves (residues 104-216 and 397-541) due to the insertion of a domain (residues 217-396, named X448 in the CAZy database) in the α5-η3 loop (Fig. 3a–c). Notably, both N-(residues 24-103) and C-terminal (residues 542-638) extensions exhibit an antiparallel seven-stranded β-sandwich fold that did not resemble any known domain at the sequence level. These two iso-structural β-sandwiches are intimately linked to the esterase core, forming a monolithic structure (Fig. 3b, c). Such structural architecture diverges from carbohydrate esterase (CE) families described in the CAZy database so far.

The active site encompasses the classical catalytic triad (Asp-His-Ser) (Fig. 3a, Supplementary Figs. 10a and 11), reminiscent from proteases[28,29]. In addition, it conserves the electropositive oxyanion hole (Supplementary Fig. 10a–c), an ancestral and recurrent feature of enzymes from the GDSL and GDSL-like families of esterases and lipases[30,31]. Interestingly, the catalytic triad is imprinted on a flat surface (Fig. 3c, and Supplementary Fig. 10b) that is uncommon in other known CE families described in the CAZy database[20] (Supplementary Fig. 10d–h). This observation agrees with a lack of selectivity to *O*-acetylated simple sugars since it does not seem to impose steric penalties to any C5 or C6 mono- and di-saccharides. Although both N- and C-terminal β-sandwich domains are remote to the catalytic center, it is proposed here that they might serve as an extended platform for XyGOs anchoring during acetate removal. Moreover, the internal X448 domain, which was not observed in the crystallographic structure, probably adds another component in the recognition mechanism of complex substrates by this new type of carbohydrate acetylesterase. These functional and structural results allow the classification of *Xac*XaeA as the founding member of the CE20 family.

**Xanthomonas enzymatic cascade for XyGOs breakdown.** The α-1,2-L-fucosidase activity (EC number 3.2.1.63) was biochemically observed in a GH95 member of the XyGUL (XAC1774), named here *Xac*Afc95. Despite more than one activity has been reported for this family, *Xac*Afc95 appeared to be very specific to L-fucose, which is equivalent to 6-deoxy-L-galactose (Supplementary Tables 6–8, and Supplementary Fig. 12h). Its crystal structure (Supplementary Table 4) conserves the canonical domain architecture of the GH95 family that consists of an N-terminal β-supersandwich (residues 36-278), α-toroidal six-hairpin catalytic domain (residues 349-703), and a C-terminal β-sandwich (residues 704-790) (Fig. 4a). The catalytic domain is connected to the super sandwich domain by a helix-rich linker (residues 279-348) and this multi-domain protomer forms dimers *in solution* (Supplementary Figs. 13j–l and 14, and Supplementary Table 10). Structural comparisons indicate that the general bases N397/N399 (carboxylate-activated) and the general acid D690 are conserved in relation to the only two structures available for this family, the α-L-galactosidase from *B. ovatus*[32] (BACOVA_03438, PDB ID 4UFC, SeqID 42.88%) and the α-1,2-L-fucosidase from *Bifidobacterium bifidum*[33] (*Bb*AfcA, PDB ID 2EAB, SeqID 30.36%) (Fig. 4b).

Previous comparisons between BACOVA_03438 (α-L-galactosidase) and *Bb*AfcA (α-1,2-L-fucosidase) led to the suggestion that the only polymorphic position at the -1 subsite would confer specificity to L-galactose or L-fucose, although the authors also pointed that a conclusive inference of the functional relevance of this polymorphic residue is hindered by the very limited structural data available for this family so far[32]. Based on structural analyses, they proposed that the presence of threonine at this position would allow a hydrogen bond with the L-galactose O6 atom, whereas a histidine would contribute to aliphatic interactions with the L-fucose C6 methyl group. However, *Xac*Afc95, which shares nearly 40% sequence identity with characterized GH95 α-1,2-L-fucosidases involved in XyG depolymerization[12,20], contains a threonine at the referred position and showed high specificity to L-fucose, contraposing the initial role proposed for this residue as a determinant for L-galactose preference (Fig. 4b, Supplementary Tables 6–8, and Supplementary Fig. 12h). In addition, the mutation T395H did not result in any change of substrate preference, supporting a less relevant role of this polymorphic position for specificity in the GH95 family (Supplementary Fig. 15a). Besides *Xac*Afc95, another characterized GH95 α-1,2-L-fucosidase (Blon_2335)[34] conserves a threonine at this polymorphic position, corroborating this hypothesis (Supplementary Fig. 16). These findings point to a more elaborate mechanism of substrate selectivity in the GH95 family that is not limited to direct interactions with the residues forming the -1 subsite.

The subsequent substitution to be cleaved after the α-1,2-L-fucoside removal is a β-1,2-galactosyl moiety. According to biochemical characterization assays, this step is performed by a β-galactosidase belonging to the GH35 family (XAC1772) with functional dependence to oligomerization. This enzyme is orthologous to GalD from *X. campestris* pv. *campestris*[35] and forms tetramers in solution (Supplementary Fig. 17a). Structural analysis revealed that the oligomerization interface involves the catalytic domain (TIM-barrel fold, residues 75–458) of one protomer and the accessory β-sandwich domain (residues 459-585) from the other subunit (Fig. 4c, and Supplementary Fig. 18). Since the functional relevance of oligomerization has not been investigated for this family so far, mutations were designed to address this question. The insertion of an arginine residue (S106R mutation) at the oligomeric interface resulted in stable monomers in solution (Supplementary Fig. 17b), which were devoid of catalytic activity (Supplementary Fig. 15b), supporting a critical role of tetramerization to the function of *Xanthomonas* GH35 members.

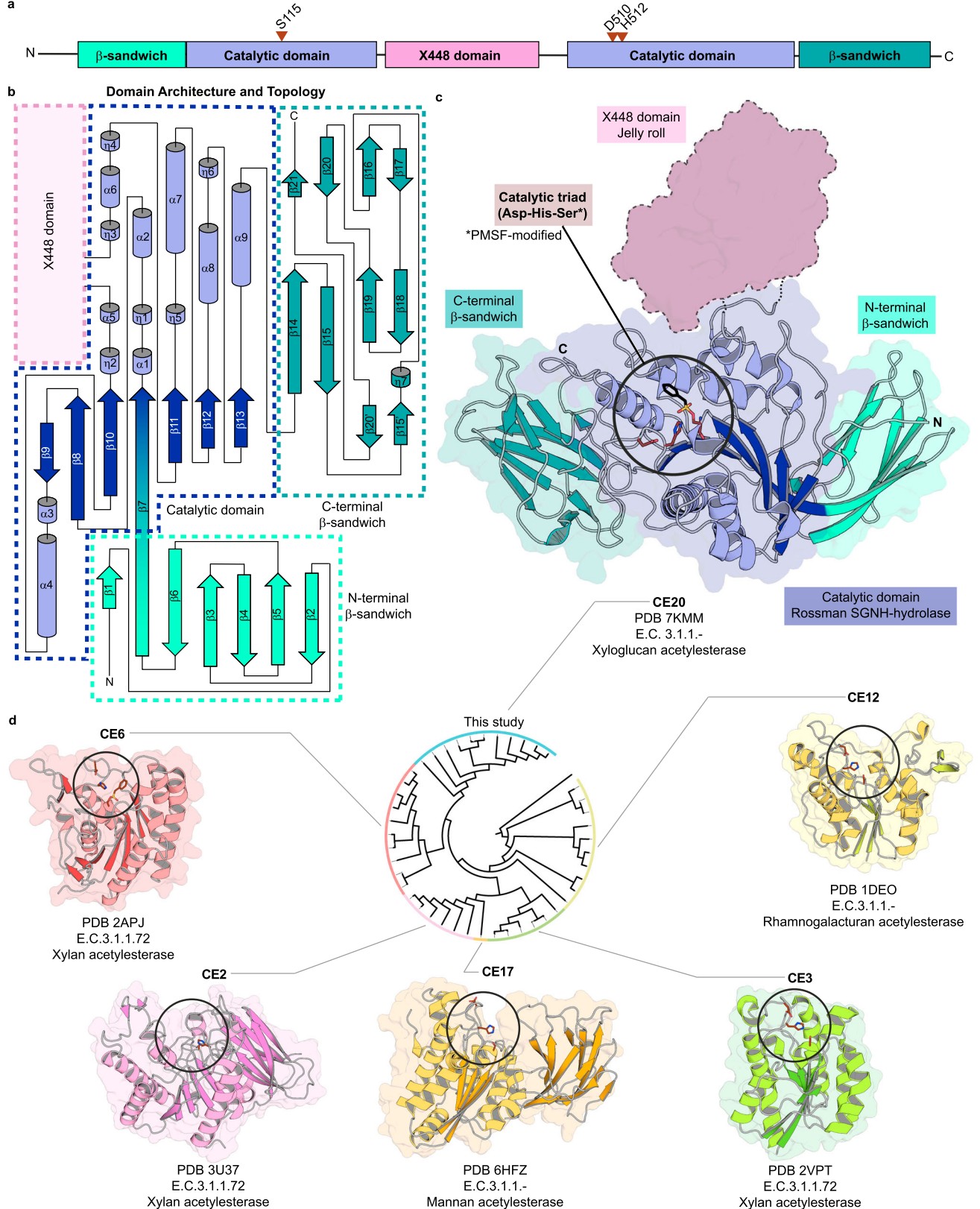

**Fig. 3 A distinctive type of carbohydrate acetylesterase. a** Domain organization showing the position of catalytic residues (red triangles); **b** structural topology and **c** crystal structure color-coded according to **a**. **d** Dendrogram of CE families based on phylogenetic analysis of the catalytic domain showing the structure of a representative member of each family. Circles indicate the active site. Catalytic residues (carbon atoms in brown) and PMSF (carbon atoms in black) are shown as sticks. The *Xanthomonas* acetylesterase, discovered in this study, is the founding member of the CE20 family.

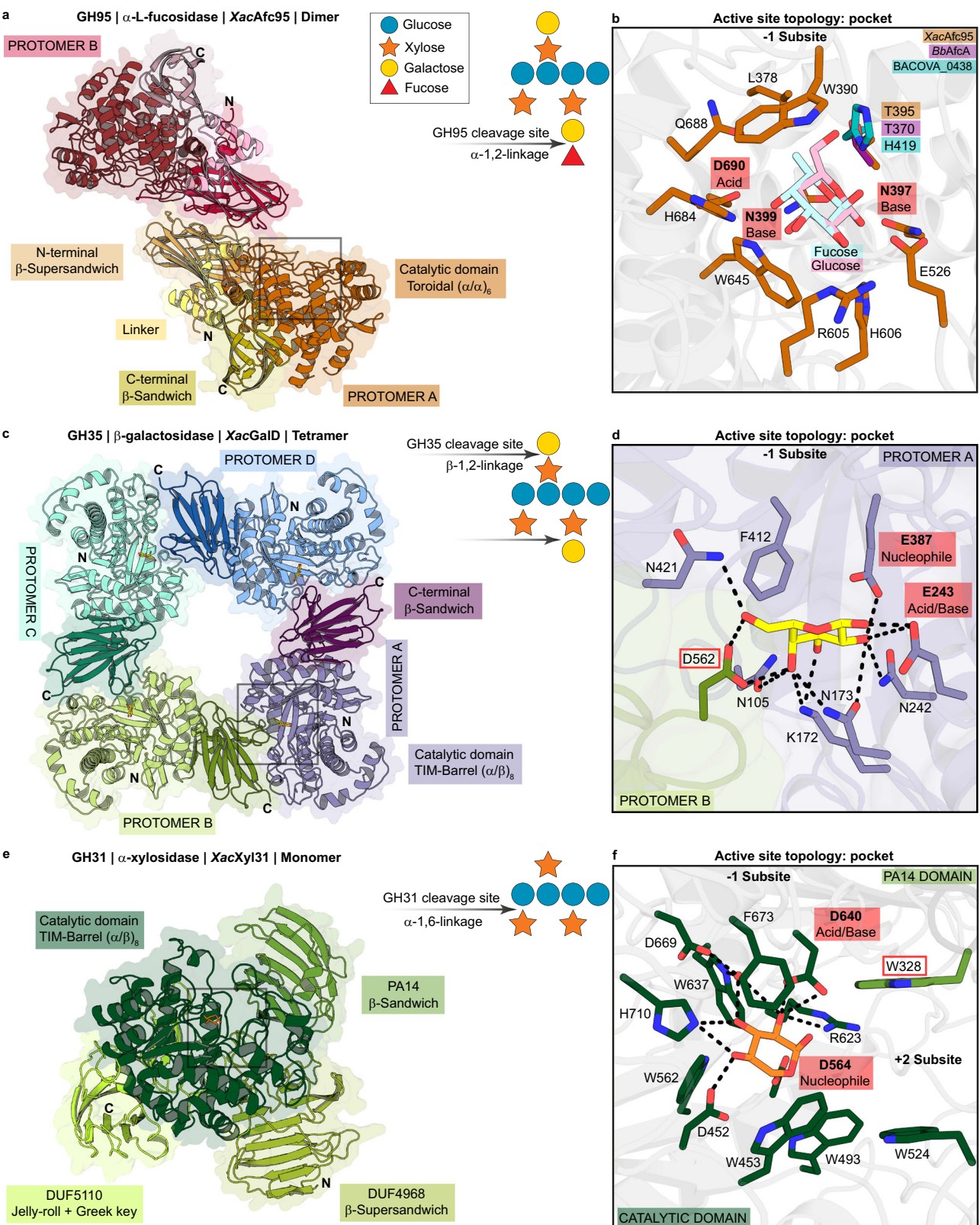

**Fig. 4 Molecular basis for the enzymatic cleavage of XyGOs decorations in *Xanthomonas*. a**, **c**, **e** Crystallographic structures of the enzymes that release the substitutions of XyGOs (diagram where blue circle = glucose, orange star = xylose, yellow circle = galactose and red triangle = fucose), highlighting their domains in shades of a specific color defining each protomer or monomer as indicated in the figure. **b**, **d**, **f** Amplified view of the enzymes active sites indicated with boxes in panels a, c, e. Residues from the -1 subsite and ligands are shown as sticks (Glucose = pink carbon atoms, Fucose = blue carbon atoms, Galactose = yellow carbon atoms and Xylose=orange carbon atoms). The red outline indicates residues selected for mutational analysis. Catalytic residues are indicated (pink boxes).

The crystal structure of *Xac*GalD with D-galactose (Supplementary Fig. 19a, b, and Supplementary Tables 4 and 5) revealed that residues in the oligomerization interface also interacts with the saccharide (Fig. 4d), demonstrating that the active-site pocket is completed by residues from the vicinal protomer. Alanine mutation of the residue D562 from the β-sandwich domain from the neighboring protomer, which interacts directly with the D-galactose in the active-site pocket (D562A), drastically reduces the enzyme substrate affinity (Supplementary Fig. 15b). Interface analysis of structurally characterized members of the GH35 family, including from archaea, eukaryota, and distinct bacterial phyla (Bacteroidetes, Proteobacteria, and Firmicutes) points out that functionalization by tetramerization is a conserved feature across the phylum Proteobacteria (Supplementary Table 11, and Supplementary Fig. 20). Besides *Xanthomonas* GH35 enzymes, other representatives from this phylum in the CAZy database, such as CC0788 from *Caulobacter vibrioides* and *Cj*Bgl35A from *Cellvibrio japonicus*, also form stable tetramers in a similar fashion as *Xac*GalD, which contrasts to the quaternary structures observed in other phyla (Supplementary Table 11).

The next step in the cascade of XyGOs processing involves an α-xylosidase from the GH31 family, according to biochemical analysis of the enzyme encoded by XAC1773 (Supplementary Tables 6–8), named here *Xac*Xyl31. The crystal structure of this enzyme (Supplementary Tables 4 and 5) revealed a conserved active-site pocket compared to other bacterial α-xylosidases from this family, including the catalytic residues (D564 as nucleophile and D640 as acid/base) and the aromatic residue (W453). The latter introduces a steric barrier to C6 saccharides, favoring only the accommodation of C5 sugars such as xylose at the -1 subsite[36] (Fig. 4e, f and Supplementary Fig. 19c, d).

Another conserved feature of GH31 α-xylosidases specific to XyGOs, present in *Xac*Xyl31, is the four-domain arrangement consisting of a central TIM-barrel catalytic domain (residues 392-757) and three all-β fold accessory domains (DUF4968, residues 38-219 and 372-391; DUF5110, residues 758-957; and PA14, residues 220-371)[36,37] (Fig. 4e). These domains comprise a monolithic tertiary structure and do not establish oligomeric contacts in any of the GH31 members structurally characterized so far, including the enzyme reported here (Supplementary Fig. 13g–i, and Supplementary Table 10). Among the four accessory domains, the PA14 is the only participating in the active site interface (Fig. 4e, f)[37–39]. This domain introduces an aromatic platform (W328) at the +2 subsite that along with W524 from the TIM-barrel catalytic domain establishes stacking interactions with XyGOs backbone (Fig. 4f)[37,40,41]. The residue W328 is highly conserved in GH31 α-xylosidases and the mutation W328A was detrimental to the catalytic activity (Supplementary Fig. 15c), supporting the importance of this region for α-xylosidase activity in the GH31 family.

As previously described in *C. japonicus*, GH31 α-xylosidases active on XyGOs may cleave specifically α(1,6)-Xyl*p* moieties appended to the non-reducing end of the backbone, requiring the cooperation of XyGOs-specific β-glucosidases to complete XyGOs depolymerization[37,42]. *Xac*Xyl31 also presented this specificity (Supplementary Fig. 21), but no β-glucosidase-encoding gene was found in the *Xanthomonas* XyGUL.

On the other hand, five potential β-glucosidases are present in the *X. citri* genome, all belonging to the polyspecific GH3 family. The heterologous expression and biochemical characterization of these enzymes revealed that two of them are β-xylosidases (*Xac*Xyl3A—XAC3076 and *Xac*Xyl3B—XAC4231) and the other three are β-glucosidases (*Xac*Bgl3A—XAC1448, *Xac*Bgl3B—XAC1793, and *Xac*Bgl3C—XAC3869) (Supplementary Tables 6 and 7).

These three β-glucosidases were expressed in the presence of XyGOs with higher expression levels of *Xac*Bgl3C and *Xac*Bgl3B

compared to *Xac*Bgl3A (Supplementary Table 12) (see section below). In addition, all three enzymes were capable of releasing the non-reducing glucosyl moiety from the XyG-derived oligosaccharides GXXG and GXG, which are the products of *Xac*Xyl31 using XXXG and XXG as substrates, respectively (Supplementary Figs. 21 and 22).

*Xac*Bgl3B displayed the highest activity on GXXG and GXG substrates and is predicted to be an outer membrane associated protein, probably exposed to the periplasm (Supplementary Fig. 22, and Supplementary Table 1). *Xac*Bgl3A, although predicted to be periplasmic, seems to be more specific to cleave β-1,3-glucooligosaccharides instead of XyGOs-derived β-1,4-glucooligosaccharides (Supplementary Figs. 23 and 24, and Supplementary Tables 1 and 8). *Xac*Bgl3C displayed a more generalist substrate profile and is predicted to be cytoplasmic, indicating that it might support the final steps of glucooligo-saccharides cleavage coming from different sources (Supplementary Figs. 23 and 24, and Supplementary Tables 1 and 8).

Together, these findings point to *Xac*Bgl3B as being the major β-glucosidase to alternate with *Xac*Xyl31 on the breakdown of XyGOs intermediates until reaching the final substrate β-1,4-glucobiose. At this last step, *Xac*Bgl3B seems also to play an important role, since it was the most efficient on β-1,4-glucobiose cleavage, compared to *Xac*Bgl3A and *Xac*Bgl3C (Supplementary Fig. 24, and Supplementary Table 8).

## XyG depolymerization products activate the expression of virulence factors.

The presence of XyGULs in both mesophyllic and vascular *Xanthomonas* pathogens indicates that XyG processing might play a role in bacterial colonization, pathogenicity and/or survival in host plants. To explore this hypothesis, the global gene expression profile of *X. citri* was assessed in the presence of XyGOs, which resulted in 276 differentially expressed genes among its 4281 protein coding sequences (Fig. 5a, and Supplementary Data 1).

As expected, the presence of XyGOs in the medium increased the expression of XyGUL genes (XAC1768-XAC1774), downstream genes related to xylose metabolism (XAC1775 and XAC1776) and an MFS transporter (XAC1777), supporting the relevance of this system for XyG processing in vivo (Fig. 5b). Moreover, these oligosaccharides upregulated two conserved GH43 encoding genes (XAC1275 and XAC4183) that are not in the vicinity of the XyGUL (Supplementary Fig. 2 and Supplementary Data 1). The enzyme encoded by XAC4183, named here *Xac*Abf43A, belongs to an underexplored subfamily, the GH43_9, with only one reported member characterized so far with a weak arabinofuranosidase activity[43]. To evaluate this activity in the *Xanthomonas* member of this subfamily, *Xac*Abf43A was produced, and biochemical assays confirmed its α-L-arabinofur-anosidase activity (Supplementary Tables 6 and 7). The recombinant production of XAC1275 did not yield a soluble and stable protein; however, it displays 37% of sequence identity with the GH43_12 arabinofuranosidase from the *B. ovatus* XyGUL[40], pointing to a potential similar functional specificity. The fact that these GH43 genes are conserved across *Xanthomonas* bacteria, positively regulated by XyGOs and functionally related to orthologues present in *Bacteroides* XyGUL indicate that *Xanthomonas* could act on both types of XyGs, fucogalactoxyloglucans (present in plants from the Rosids clade[44]) and arabinoxyloglucans (present in plants from the Asterids clade, in the Solanaceae and Oleaceae families[45]).

The sensing of XyG depolymerization products by *Xanthomonas* also modulated other aspects of the bacterial metabolism (Fig. 5c). While repressing genes related to chemotaxis and flagellar motility, XyGOs stimulated processes that play essential

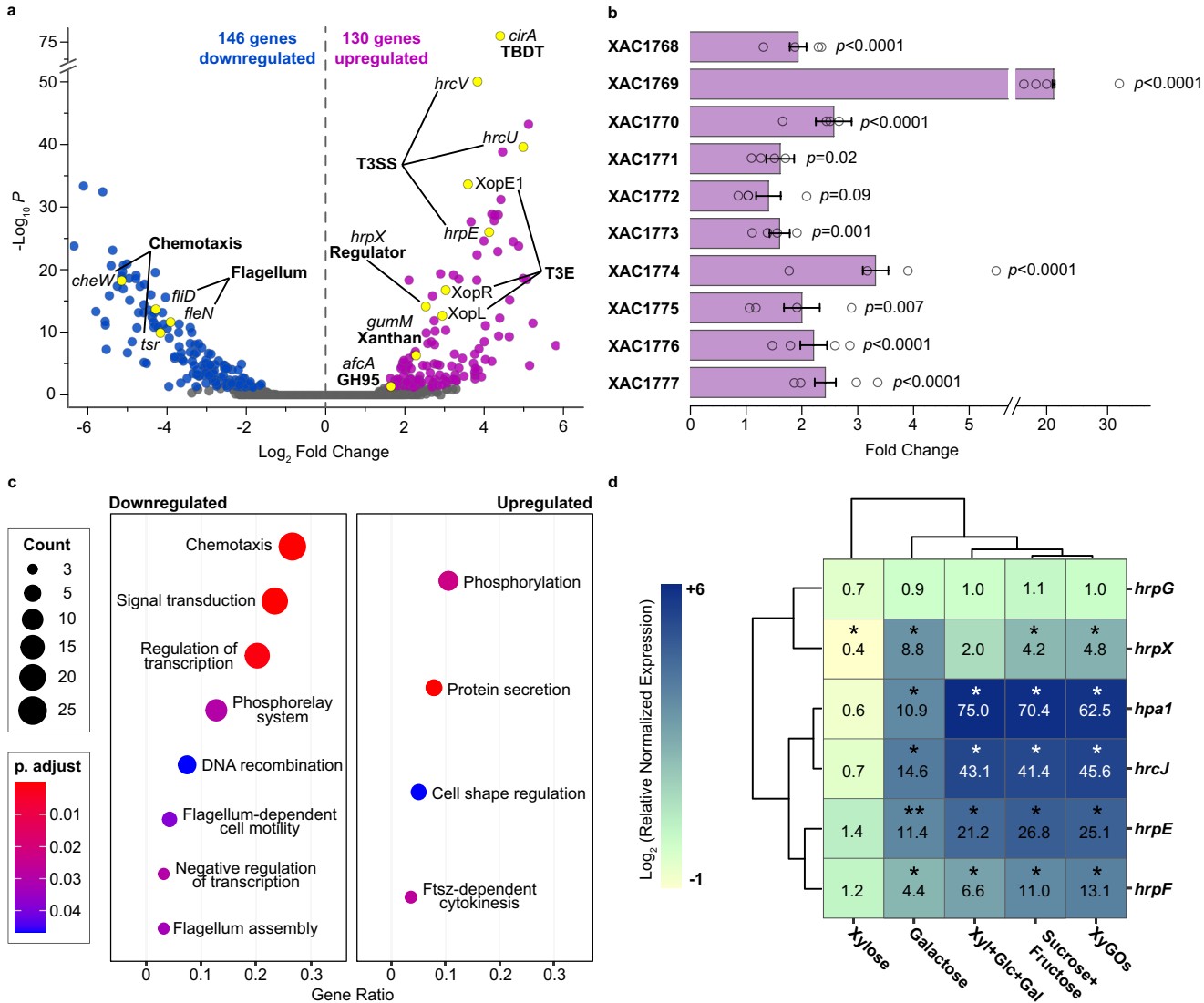

**Fig. 5 Transcriptional responses to XyGOs in *X. citri*. a** Volcano plot of RNA-seq data highlighting differentially expressed genes (DEGs; blue circles = downregulated and purple circles = upregulated). In yellow are DEGs acting on processes important for virulence or XyG utilization. Genes were considered differentially expressed according to Wald test implemented in DESeq2. *p*-values were adjusted for multiple tests using the Benjamini-Hochberg (BH) method implemented in DESeq2. Thresholds: *p*-adjusted < 0.05 and | log₂ Fold Change| > 1. **b** Transcription levels of XyGUL genes, downstream genes encoding xylose metabolism and MFS transporter in RNA-seq experiments. Data shown as mean ± SE from four biologically independent experiments (*n* = 4) (open circles). Threshold: BH adjusted *p*-value < 0.05. **c** Gene ontology (GO) enrichment analysis of DEGs. Circles size and color represent the counts and BH adjusted *p*-values, respectively, according to the legend at the left. Gene ratio is the number of DEGs related to a GO term divided by the total number of annotated DEGs. Categories were considered enriched based on hypergeometric test, implemented in the clusterProfiler 3.14.3R/Bioconductor enrich function[60]. **d** Clustered heat map of RT-qPCR data of *hrp* genes color-coded according to the log₂ (−1 to +6) of the mean normalized expression relative to the control (numbers inside the boxes). Two-tailed *t* test: *, *p* < 0.001; **; *p* < 0.0001; Xyl = xylose, Glc = glucose and Gal = galactose. See details in Supplementary Fig. 25.

roles in the early stages of plant infection[46], including xanthan gum biosynthesis and secretion of type III protein effectors into the host cells. Seven *gum* genes (XAC2574-XAC2580), the entire T3SS cluster (XAC0393-XAC0417), and 14 effector protein genes were up-regulated by XyGOs (Supplementary Data 1, and Supplementary Table 13), indicating an unprecedented role of XyG depolymerization products in bacterial virulence. As XyGs have a complex structure, RT-qPCR assays were performed to assess whether the activation of T3SS genes depends on the structure of XyGOs or its basic core constituents, i.e., glucose, galactose, and xylose. In these experiments, the mixed-sugar condition (glucose plus galactose and xylose) activated the expression of T3SS genes akin to the XyGOs condition, showing

that the signaling effect of XyGOs relies on its monomers and not on its complex structure (Fig. 5d, and Supplementary Fig. 25).

To gather insight into the signaling pathway assessed by XyG depolymerization products, we also evaluated the expression of master regulators of T3SS expression, *hrpG* and *hrpX*[47]. Although *hrpG* was not activated in any of the tested conditions, the expression of *hrpX* increased in the presence of galactose, XyGOs and sugar mix, being maximal in the galactose condition (Fig. 5d, and Supplementary Fig. 25). This result shows that galactose is sufficient for the transcriptional activation of *hrpX*. However, for all T3SS genes analyzed, the transcription was further stimulated when galactose was supplemented with glucose and xylose, indicating that a second signal coming from other XyG

components boosts the expression of T3SS genes (Fig. 5d, and Supplementary Fig. 25).

The combination of the monosaccharides mimicking the XyG breakdown products can also be achieved by the synchronized depolymerization of other plant polysaccharides and is as potent as the combination of sucrose and fructose in stimulating the T3SS expression in vitro (Supplementary Fig. 25), supporting that *Xanthomonas* relies on multiple and redundant sources of signals to trigger virulence and modulate host responses. Therefore, to confirm this redundancy hypothesis, we evaluated whether *X. citri* pv. *citri* would maintain its virulence even with the knockout of XyGUL genes (TBDTs and xyloglucanase) or an adjacent MFS transporter (Supplementary Fig. 26). As envisaged, the wild-type phenotype was preserved in these mutants, supporting that the activation of T3SS by XyG depolymerization products is probably compensated via functional redundancy or via alternative pathways for virulence activation, especially in the case of ΔXAC1768-69 deletion. The in vitro growth of this mutant in minimal medium containing XyGOs was severely reduced, indicating that the lack of XAC1768-69 genes impairs the uptake of these oligosaccharides (Supplementary Fig. 5a).

The deletion of the inner-membrane MFS transporter gene (XAC1777) did not affect the bacterial growth using either XyGOs or its basic components as primary carbon source, supporting that at least one of the other 40 MFS transporters encoded by *X. citri* genome compensate its absence (Supplementary Fig. 5, and Supplementary Table 14). Notably, the strain lacking the GH74 xyloglucanase (ΔXAC1770) displayed a XyG depolymerization halo similar to the wild-type strain, suggesting that another endo-β-1,4-glucanase would be functionally redundant to the XyGUL GH74 enzyme (Supplementary Fig. 27). A search for endo-β-1,4-glucanases in *X. citri* genome resulted in one GH9 member (XAC2522, *Xac*Egl9), one GH8 member (XAC3516, *Xac*Cel8[48]), and five putative GH5 glucanases (XAC0612—subfamily GH5_1, *Xac*EngXCA; XAC0028—subfamily GH5_5, *Xac*Egl5A; XAC0029—subfamily GH5_5, *Xac*Egl5B; XAC0030—subfamily GH5_5, *Xac*Egl5C; and XAC0346—not yet assigned to a subfamily)[49]. The recombinant production and activity assays of these enzymes revealed that only *Xac*Egl9, *Xac*Egl5B, and *Xac*EngXCA are able to cleave XyG, with *Xac*Egl9 showing the highest specific activity on this polysaccharide (Supplementary Table 15). This result indicates that these glucanases might compensate the absence of the GH74 xyloglucanase in the ΔXAC1770 mutant. Characterization of *Xac*Egl9 revealed kinetics parameters on XyG akin to those found for *Xac*Xeg74 (Supplementary Table 8, and Supplementary Figs. 6g and 12i), supporting the role of other endo-β-1,4-glucanases in the XyG cleavage in strains lacking the GH74 enzyme.

## Discussion

Here, we show that most *Xanthomonas* species, a highly diverse bacterial genus that infects hundreds of plants, have an intricate enzymatic toolbox to break down XyGs. The *Xanthomonas* XyGUL encodes oligomeric and multi-modular glycoside hydrolases (GH74 xyloglucanase, GH31 α-xylosidase, GH35 β-galactosidase and GH95 α-L-fucosidase) and a distinguishing carbohydrate acetylesterase with no significant similarity with known CAZy families and not present in any similar XyGUL characterized so far[5,6,12] (Figs. 1 and 6). This novel acetylesterase, *Xac*XaeA, is the founding member of the CE20 family.

Transcriptomic, biochemical and gene deletion analyses revealed that the *Xanthomonas* XyGUL is complemented by other conserved CAZyme genes encoding GH9 and GH5 enzymes with xyloglucanase activity, GH3 β-glucosidases and GH43 arabinofuranosidases. The GH9 (*Xac*Egl9) and GH5 (*Xac*Egl5B) enzymes serve as redundancy components to the pivotal xyloglucanase

activity that initiate XyG breakdown and the GH43 enzymes, such as *Xac*Abf43A, confer the ability to these bacteria to cleave both fucogalactoxyloglucans[44] and arabinoxyloglucans[45]. In the final steps of XyG processing, the GH31 α-xylosidase (*Xac*Xyl31) acts coordinately with a XyGOs-active β-1,4-glucosidase (*Xac*Bgl3B) at the non-reducing end of XyGOs, cycling between xylosyl side chain cleavage and glucosyl main chain removal, as previously observed in the saprophyte *C. japonicus*[5,42]. A difference is that *C. japonicus* displays a β-glucosidase with the highest efficiency toward the intermediate GXXG and another one with the highest efficiency on β-1,4-glucobiose, whereas in *X. citri* the enzyme *Xac*Bgl3B seems to play both roles. These observations highlight that microbial systems, according to their ecological niches, have evolved singular and equally complex molecular strategies to cope with the structural and chemical diversity of XyGs.

From a mechanistic point of view, the enzymes encoded by *Xanthomonas* XyGUL also show distinguishing properties compared to other characterized homologs. The GH74 enzyme harbors an unusual arginine-based mechanism of substrate recognition, contrasting to the canonical strategy in the GH families based on aromatic CH–π interactions[50] (Fig. 2). The xyloglucan acetylesterase features an unprecedented molecular architecture with two iso-β domains at both N- and C-termini and the insertion of an uncharacterized domain in the middle of the catalytic core (Fig. 3). The GH95 member demonstrates that substrate specificity in this family does not involve polymorphic positions in the -1 subsite, pointing to the relevance of indirect interactions in determining selectivity. In addition, the role of ancillary domains and oligomerization is prominent in the function of *Xanthomonas* XyGUL enzymes (Fig. 4).

Notably, the XyG depolymerization products play a role in the activation of multiple genes related to virulence in *Xanthomonas* spp., including those encoding effector proteins and the T3SS that inject these virulence factors into the host cells (Fig. 5). In the genus *Xanthomonas*, the activation of T3SS expression by carbohydrates such as sucrose and fructose in *X. campestris* pv. *campestris* and *X. campestris* pv. *vesicatoria*[51,52], and xylose in *X. oryzae* pv. *oryzae*[53–55], has been previously demonstrated. However, the role of galactose in this process and XyG depolymerization as a source of T3SS inducers, are novel components in the complex regulatory mechanisms of virulence in these pathogens.

Our results demonstrate that galactose from XyG depolymerization activates the transcription of *hrpX*, a master regulator of T3SS expression (Fig. 6). The role of galactose as an inducer of *hrpX* gene correlates with a previous study in *X. oryzae* showing that activation of *hrpX* expression is mediated by a regulator of galactose metabolism, termed GamR[56] (XAC1767). In addition to galactose activation, a second signal from xylose, acting after the *hrpX* expression, likely contributes to the expression of T3SS genes. Similarly to that proposed for *X. oryzae*, the induction of T3SS by xylose might be associated with a post-transcriptional mechanism that suppresses HrpX proteolysis[54]. Based on these observations, we propose that during XyG depolymerization, the released galactose activates the transcription of *hrpX*, likely by modulating the GamR activity, whereas the released xylose suppresses HrpX degradation, thus promoting the activation of T3SS genes and other HrpX-mediated processes (Fig. 6). It is noteworthy that most *Xanthomonas* bacteria are equipped with other CAZymes and polysaccharide utilization loci specialized in depolymerizing other hemicelluloses[57,58] and pectins[59], which can also generate xylose and galactose, adding more layers of complexity in the modulation of virulence and pathogenesis mediated by host carbohydrate processing in these phytopathogens.

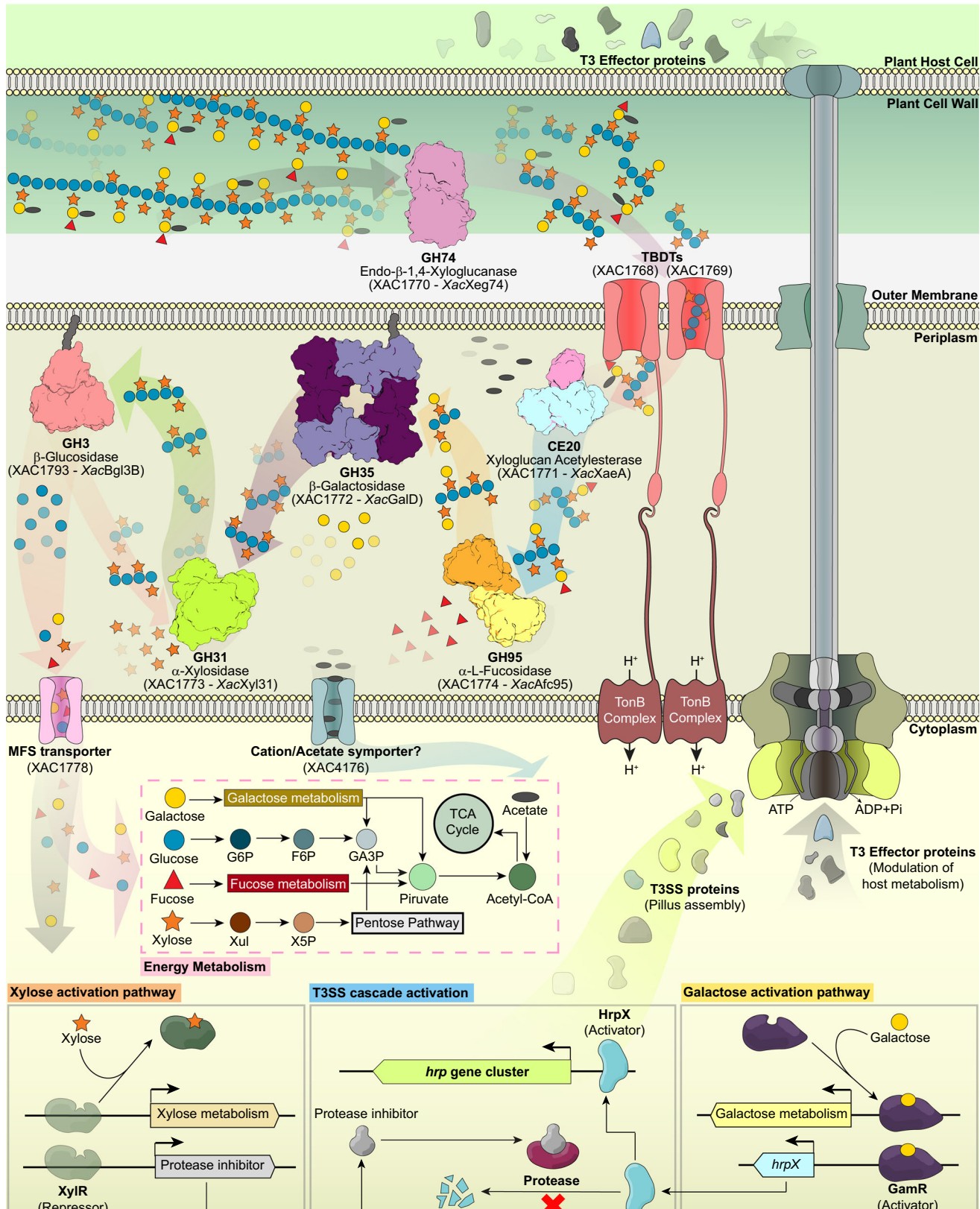

**Fig. 6 XyG enzymatic breakdown and the role of released products in the metabolism and virulence of *X. citri*.** Representation of the pathogen-host cell interface highlighting how XyG is progressively depolymerized by *Xanthomonas* enzymes and internalized by transmembrane transporters. In this bacterium, monosaccharides released from XyG induce the expression of multiple genes, including the T3SS machinery via mechanisms suggested based on transcriptional data and literature[54–56] (details in Supplementary Fig. 28, and Supplementary Table 1). T3, Type III; T3SS, Type III secretion system; TBDTs, TonB-dependent transporters; MFS, major facilitator superfamily transporter; G6P, D-glucose-6-phosphate; F6P, D-fructose-6-phosphate, GA3P, D-glyceraldehyde 3-phosphate; Xul, D-xylulose; X5P, D-xylulose 5-phosphate; and TCA, Tricarboxylic Acid cycle.

In conclusion, this work provides an in-depth understanding of the multi-enzymatic system employed by plant-associated bacteria for XyG depolymerization, and also establishes a novel component in the regulatory mechanisms of virulence and pathogenesis in *Xanthomonas*.

## Methods

**Phylogenetic analysis**. Phylogenetic analysis of *Xanthomonas* species was performed based on a set of 92 single-copy core genes according to the UBCG pipeline 3.0[61]. Individual proteins were aligned separately using MAFFT 7.299b and then concatenated. Phylogenetic analysis was inferred using RAxML 8.2.0 and the PROTGAMMAWAG model with 1,000 bootstrap replicates. Applying a similar protocol, phylogenetic analysis of carbohydrate acetylesterases was perfomed with sequences of the catalytic domain of characterized enzymes available at the CAZy database (http://www.cazy.org/) including homologous sequences of *Xac*XaeA.

**Molecular cloning and site-directed mutagenesis**. The nucleotide sequence encoding the XyGUL and accessory enzymes were amplified from the genomic DNA of *X. citri* pv. *citri* 306 strain or *X. campestris* pv. *campestris* ATCC 33913 using standard methods (Supplementary Table 16) and confirmed by Sanger sequencing. PCR-amplified gene fragments were cloned into the pET28a and pETM11 vectors. The mutants were prepared using the QuikChange II XL Site-Directed Mutagenesis Kit (Agilent) (Supplementary Table 17).

**Protein expression and purification**. The proteins were expressed in *Escherichia coli* strains as described in the Supplementary Table 18. Proteins were purified by two chromatographic steps as detailed in the Supplementary Table 19. Purified samples were analyzed by SDS-PAGE and dynamic light scattering (DLS) in a Malvern ZetaSizer Nano series Nano-ZS (model ZEN3600) instrument (Malvern Zetasizer). DLS data were collected and analyzed with Zetasizer (7.12) software to evaluate sample homogeneity.

**Analytical hydrodynamic analysis**. Size-exclusion chromatography coupled with multi-angle light scattering (SEC-MALS) experiments were performed at 25 °C using a triple-angle static light scattering detector miniDAWN™ TREOS and Optilab® T-rEX refractive index monitor (Wyatt Technology) coupled to an ÄKTA fast protein liquid chromatography system (GE Healthcare) with a Superdex 200 HR 10/300 GL analytical size-exclusion column (GE Healthcare)[62]. 250 μL from purified wild-type *Xac*GalD and mutant S106R at 50 μmol L$^{-1}$ were injected separately into the column and eluted in 20 mmol L$^{-1}$ HEPES pH 7.5, 150 mmol L$^{-1}$ NaCl. Data were processed using the ASTRA V software 6.0 (Wyatt Technology).

Small-angle X-ray scattering (SAXS) data collection was performed with protein samples at different concentrations (1, 3, 5, and 10 mg mL$^{-1}$) at the D01A-SAXS2 beamline at the Brazilian Synchrotron Light Laboratory (LNLS-CNPEM, Campinas, Brazil), using a CCD-Mar165 detector and fit2D software v. 18. Data were processed and analyzed with the ATSAS package 4.8.6[63] using the programs GNOM v. 5.0, DAMMIN v. 5.3, DAMAVER v. 5.0, CRYSOL v. 2.8.3, and SUPCOMB v. 2.3. SAXSMoW server[64] was used for protein molecular weight determination, and oligomeric interface interaction energy was calculated using the PDBePISA server[65].

**Protein crystallization, X-ray data collection, and structure determination**. Proteins were crystallized by the vapor diffusion method (Supplementary Table 20). Diffraction data were acquired under cryogenic conditions at the MX2 beamline from the Brazilian Synchrotron Light Laboratory (LNLS-CNPEM, Campinas, Brazil) using a Pilatus 2M detector (Dectris) and MXCuBE 2 (Qt4) software or at the BL9-2 beamline from the Stanford Synchrotron Radiation Lightsource (SSRL-SLAC, Menlo Park, USA) using a Pilatus 6M detector (Dectris) and BluIce 4.0 software. Data were indexed, integrated and scaled using the XDS package v. Jan 31st 2020 Built 20200417[66]. *Xcc*Xeg74, *Xac*GalD, *Xac*-Xyl31 and *Xac*Afc95 structures were solved by molecular replacement method using the PHASER software from PHENIX package dev-3139[67] and the atomic coordinates from homologous proteins 2CN2[68], 4D11[5], 2XVG[37] and 4UFC[32], respectively. Structure of *Xac*XaeA was solved by zinc SAD using the programs SHELXC/D/E from CCP4i package 7.0.023[69]. The initial model of *Xac*XaeA was obtained with AutoBuild Wizard from the PHENIX package dev-3139[70] and further refined iteratively with COOT 0.8.9[71] and PHENIX_refine dev-3139 programs. Structure validation was carried out with the Molprobity server[72]. Metal-binding sites validation was performed with the CheckMyMetal server[73–75]. Carbohydrate complexes structures were evaluated using Privateer software from CCP4i2 package 1.0.2 revision 5710[76] and figures were generated using Pymol v. 2.3 or 1.3. Data collection, processing and analyses are summarized in Supplementary Tables 4 and 5.

**Glycoside hydrolase assays**. XyGUL and accessory GHs activities were evaluated against several synthetic and natural substrates as described in Supplementary Table 21. The enzyme amount and reaction time for enzyme assays were

determined based on linearity tests previously performed. Spectrophotometric data were collected in an Infinite 200 PRO microplate reader (Tecan) using the i-Control 1.10.4.0 software (Tecan). Kinetic parameters were determined from substrate saturation curves using the OriginPro 8.1 software. All enzyme assays consist of at least three independent experiments.

*C. langsdorffii* XyG was extracted from the cotyledons powder by washing three times with 80% ethanol solution at 80 °C for 10 min to remove low molecular weight carbohydrates. Polysaccharides were extracted by resuspending in water the ethanol-insoluble portion of the material and incubating for 8 h at 80 °C under constant agitation. The mixture was filtrated, added three volumes of ethanol and centrifuged (12,000 *g* for 15 min). The supernatant was discarded and the pellet dried at 80 °C[77]. The XyGOs released from *C. langsdorffii* XyG by *Xcc*Xeg74 enzyme were analyzed by the HPAEC-PAD system (Dionex) equipped with the CarboPac PA100 analytical column (Dionex). Identification of XyGOs was performed by MALDI-TOF on a Bruker Autoflex MALDI-TOF mass spectrometer (Bruker Daltonics) in reflectron positive mode with a 19 kV voltage and covered mass within the *m/z* values of 700–3500.

The activity of the enzymes *Xac*BglA, *Xac*BglB, and *Xac*BglC on XyGOs was investigated after the removal of the xylosyl moiety at the non-reducing end glucosyl residue in XyGOs by the β-xylosidase *Xac*Xyl31. Reactions with *Xac*Xyl31 (0.1 mg mL$^{-1}$) were performed using 10 mg mL$^{-1}$ XyGOs (Megazyme O-X3G4) incubated at 45 °C and pH 6.5. After 60 min, the reactions were stopped by boiling for 5 min and the final products (GXXG and GXG) were used for activity assays with the three β-glucosidases. *Xac*BglA, *Xac*BglB, and *Xac*BglC reactions consisted of adding 0.1 mg mL$^{-1}$ of each enzyme to the solution of GXXG and GXG. The temperatures chosen in the reactions were 45 °C (*Xac*BglA), 25 °C (*Xac*BglB) or 35 °C (*Xac*BglC). Samples were collected at 0, 30 min, 1, 2, 3, and 24 h and reactions were stopped by the addition of methanol. A total of 5 μL of the quenched reactions were added to 95 μL of 5 μmol L$^{-1}$ xylohexaose (used as the internal standard) in water and injected into an LTQ XL TM linear ion trap mass spectrometer (Thermo Fisher Scientific) in scan mode (*m/z* 300–1300). Samples were directly infused at a rate of 10 μL min$^{-1}$ into an ESI(+) source with a spray voltage maintained at 4.0 kV and heated to 250 °C in the source.

The xyloglucanase activity of wild-type and mutant *Xanthomonas* strains were monitored in plate assays. Cultures were grown overnight in LBON medium (1% *m/v* bacto peptone and 0.5% *m/v* yeast extract) at 30 °C and 200 rpm, diluted to OD$_{600 nm}$ 0.4, plated (0.5 μL) on solid NYG medium (5 g L$^{-1}$ peptone, 3 g L$^{-1}$ yeast extract, 20 g L$^{-1}$ glycerol, 15 g L$^{-1}$ agar) supplemented with 0.5% of tamarind xyloglucan (Megazyme), and grown for 40 h at 30 °C. Activity halos were revealed with 5 mg mL$^{-1}$ Congo red and successive washes with 1 mol L$^{-1}$ NaCl.

***Arabidopsis thaliana* cultivation and XyGOs enzymatic extraction**. *Arabidopsis thaliana* Col-0 seeds were sterilized in 70% ethanol solution for 2 min in a 2 mL tube. The supernatant was discarded and followed by addition of 10% (*v/v*) sodium hypochlorite, 10 μL Tween 20 (Sigma-Aldrich) solution for 5 min under agitation. Then, the material was washed five times with sterile water and stratified in water for 48 h at 4 °C protected from light. Seeds were plated over Murashige and Skoog[78] sterile media (Sigma-Aldrich) at half strength with 10% agar. Plates were exposed to a photosynthetic photon flux density of 200 μmol m$^{-2}$ s$^{-1}$ for 2 h and then grew in a Phytotron chamber (Fitotron HGC Weiss Technik) for 6 days in the dark at 21 °C, 70% humidity. Seedlings (around 1 cm long in height) were harvested, weighted, frozen in liquid nitrogen and stored at −80 °C[79].

Frozen seedlings were homogenized in a Retschmill (model MM200, Retsch) at 25 Hz for 1 min. The grounded plants were washed three times by resuspending it in 1 mL methanol, vortexing, centrifuging at 10,000 *g* for 10 min and discarding the supernatant. The material was dried for 5 min under vacuum and washed twice with 500 μL of water, discarding the supernatant[79]. The resulting residue was used for acetylated XyGOs extraction. A proportion of 1 μg of purified *Xac*Xeg74 enzyme for every 50 mg of starting seedlings was utilized for digesting the acetylated XyG in 200 μL of 50 mmol L$^{-1}$ ammonium formate pH 5.0 solution for 16 h at 30 °C, 450 rpm. Reaction was stopped by heating the mixture for 2 min at 95 °C. The resulting solution was centrifuged twice at 1000 *g* for 5 min, the pellet discarded and the supernatant finally stored at −20 °C upon utilization.

**Xyloglucan acetylesterase assays**. Acetylated mono- and disaccharides were chemically prepared with excess of acetic anhydride in the presence of the catalyst pyridine[80] and validated by $^1$H nuclear magnetic resonance. Spectra analysis showed that the acetylation on both monosaccharides (mannose pentaacetate, galactose pentaacetate, fucose tetraacetate, arabinose tetraacetate and xylose tetraacetate) and disaccharides (β-1,4-glucobiose octaacetate and saccharose octaacetate) was not selective. The $^1$H nuclear magnetic resonance spectra were recorded on DD2 spectrometer (Agilent) from Brazilian Biosciences National Laboratory (LNBio-CNPEM, Campinas, Brazil), operating in Larmor frequency of 499.726 MHz equipped with triple resonance probe. NMR data processing was performed using VnmrJ software (4.2 Revision A). Other used acetylated sugars were purchased including *N*-acetylglucosamine, *N*-acetylneuraminic acid (Sigma-Aldrich), α-glucose pentaacetate (Santa Cruz Biotechnology), β-galactose pentaacetate and β-glucose pentaacetate (Combi-Blocks). Reactions consisted of 0.01 mg mL$^{-1}$ of the enzyme *Xac*XaeA and 5 mmol L$^{-1}$ of each acetylated sugars incubated during 15 min at 20 °C and 600 rpm in 50 mmol L$^{-1}$ HEPES buffer pH 7.5.

Reactions were stopped by adding 40 µL of methanol. Final products and residual substrates were monitored on a Waters Synapt HDMS system at V mode, and ESI (+) with a spray voltage maintained at 3.0 kV and heated to 130 °C in the source using MassLynx 4.1 software. A volume of 15 µL of the quenched reactions and 2 µL of 1 mmol $L^{-1}$ xylotetraose (used as the internal standard) were added to 183 µL of water and injected into the mass spectrometer in scan mode ($m/z$ 150–900) with direct infusion at a flow rate of 50 µL $min^{-1}$ [81].

Esterase reactions on acetylated XyGOs extracted from *A. thaliana* (see section above) consisted of incubating 0.02 mg $mL^{-1}$ of *Xac*XaeA with the substrate (estimated concentration 5–10 µg $mL^{-1}$) for 2 and 24 h at 20 °C and 600 rpm on 50 mmol $L^{-1}$ HEPES buffer pH 7.5. 100 µL of quenched reactions were desalted using Oasis HLB cartridges (Waters). HLB cartridges were first activated with methanol and equilibrated with water according to the manufacturer's protocol. The samples were applied and then washed seven times with 1 mL water. XyGOs were eluted with 150 µL 25% ($v/v$) methanol in water. Controls, final products and residual substrates were analyzed on an LTQ XL TM linear ion trap mass spectrometer (Thermo Fisher Scientific). The samples were directly infused at a rate of 10 µL $min^{-1}$ into the ESI(+) source in scan mode ($m/z$ 150–2000). The spray voltage maintained at 4.2 kV and heated to 280 °C in the source. CID-MS/MS fragmentation analysis of XyGOs were performed using different collision energies (15–35) and the isolation window was set to 1 Th. Estimated mass/charge ratio of acetylated and non-acetylated oligosaccharides were compatible to the literature[44,79] and confirmed by MS/MS fragmentation fingerprint.

**Xanthomonas cultivations**. For growth curve analysis, *X. citri* strains was cultured in LBON medium (1% *m/v* bacto peptone and 0.5% *m/v* yeast extract) containing 100 µg $mL^{-1}$ ampicillin at 30 °C and 200 rpm until mid-exponential phase. Then, the harvested cells were washed once and transferred to the modified minimal medium XVM2[52] (XVM2m, without sucrose and fructose, containing different sugar sources at a final concentration of 5 mg $mL^{-1}$), for an initial $OD_{600 nm} = 0.01$. Growth was monitored for 30 h, at 30 °C, in a SpectraMax M3 Multi-Mode Microplate Reader (Molecular Devices). Four biological replicates were used for each condition. XyGOs used in *Xanthomonas* growth assays were prepared by incubating 5 mg $mL^{-1}$ tamarind xyloglucan (Megazyme) with *Xac*Xeg74 (4 µg $mL^{-1}$) at 30 °C for 14 h. The reaction was stopped by heating at 80 °C for 15 min.

**RNA sequencing and analysis**. Total RNA was extracted from 15 mL of *X. citri* cultures grown on XVM2m + XyGOs or XVM2m + glucose medium (see the section above) at the mid-exponential phase using the TRIzol/chloroform protocol[82]. Samples were further treated with RNase-free DNaseI (Invitrogen) and RNaseOUT (Invitrogen) and purified with the RNeasy Mini Kit (Qiagen), according to the manufacturer's recommendations. In addition, RNA integrity was evaluated in an Agilent 2100 Bioanalyzer (Agilent Technologies) and samples were quantified in a Qubit® 2.0 Fluorometer using the RNA BR assay kit (Life Technologies). Libraries were prepared according to the manufacturer's protocol of the TruSeq Stranded Total RNA kit (Illumina Inc.). Sequencing was performed on the Illumina HiSeq 2500 platform (LNBR-CNPEM, Campinas, Brazil). RNA-seq data were deposited in the Gene Expression Omnibus database under accession number GSE159288.

RNA-seq raw reads were filtered to remove low-quality reads and adapters sequences using Trimommatic v. 0.38[83] and rRNA reads were removed using SortMeRNA 2.1[84] (Supplementary Table 22). High-quality reads were mapped to the *Xanthomonas citri* pv. *citri* 306 genome[85] using Bowtie2 v.2.2.5 algorithm[86] and reproducibility among the biological replicates was assessed by the Principal Component Analysis and Pearson correlation methods. Differential expression analysis was carried out by pairwise comparison between *X. citri* grown in XVM2m containing XyGOs and XVM2m glucose medium using $|log_2$ Fold Change$| \geq 1$ and a $p$ adjusted $\leq 0.05$ as thresholds using the Bioconductor DESeq2 v.1.18.1[87] package in the R v.3.4.1 platform[88].

**RT-qPCR analysis**. RNA-seq data were analyzed for the identification of potential reference genes. The arithmetic mean of the TPM values (transcription per million reads) of each gene was calculated in all conditions, followed by the determination of the variation coefficient and MFC (ratio between the maximum and minimum TPM value of each gene)[89,90]. From the 20 potentially most stable genes, the targets XAC2293, XAC2177, XAC4047, and XAC4218 were selected for RT-qPCR experiments based on their mean TPM and $p$ values (Supplementary Table 23). The expression stability was evaluated for the potential reference genes and the Cq values (quantification cycles) were analyzed using three different statistical tools: BestKeeper[91], NormFinder[92] and RefFinder[93] (Supplementary Table 24, and Supplementary Fig. 29). RT-qPCR assays were performed in an Applied Biosystems ViiA™ 7 Real-Time equipment (Life Technologies) using the Power SYBR® Green RNA-to-CT™ 1-Step Kit (Life Technologies) as detailed in Supplementary Table 25. The relative normalized expression values for each gene were calculated according to the $2^{-\Delta\Delta Ct}$ method[94]. Data were log-transformed and statistically compared by ANOVA and unpaired 2-tailed $t$ test using Prism 8.4.1 software (GraphPad). The correlation between gene expression data obtained in RNA-seq and RT-qPCR assays can be assessed in Supplementary Fig. 30.

**Gene knockout in X. citri**. Single (ΔXAC1768, ΔXAC1769, ΔXAC1770, ΔXAC1777) and double (ΔXAC1768-XAC1769) gene knockout mutants were obtained by a two-step allelic exchange procedure. DNA fragments (~1.2 kb) corresponding to regions upstream and downstream to the target genes were amplified by PCR from the *X. citri* genome (Supplementary Table 26). Each corresponding pair of fragments was ligated and then cloned into the pNPTS138 suicide vector[95] in the corresponding restriction sites. The plasmids were introduced into *X. citri* by electroporation (~2.3 kV, ~5 ms), and sucrose-sensitive and kanamycin-resistant colonies were selected (LBON-agar, 100 µg $mL^{-1}$ ampicillin, 100 µg $mL^{-1}$ kanamycin with and without 5% sucrose, respectively). This step selected colonies that suffered the first homologous recombination event, when the plasmid is inserted into the bacterial genome. These colonies were grown in LBON, 100 µg $mL^{-1}$ ampicillin without selection to allow the occurrence of the second recombinant event, when the plasmid is excised from the genome. The cultures were plated, and individual colonies were selected for simultaneous sucrose resistance and kanamycin sensitivity. Deletions were confirmed by PCR and DNA-sequencing (Supplementary Table 27).

**Virulence assays**. Plants of sweet orange (*Citrus sinensis* 'Natal') were infiltrated by the pinprick method[96] with water suspensions of *X. citri* at $OD_{600}$ of 0.1 previously grown in LBON agar plates, supplemented with ampicillin (100 µg $mL^{-1}$), for 48 h at 28 °C. The assays were performed with three independent biological samples, each composed of 16 technical replicates. Plants were maintained under greenhouse conditions and monitored daily for the appearance of canker symptoms. Quantitative analyses of canker lesions were performed using ImageJ v. 1.53b.

**Reporting summary**. Further information on research design is available in the Nature Research Reporting Summary linked to this article.

## Data availability
Atomic coordinates and structure factors have been deposited in the Protein Data Bank (PDB) with accession codes 7KN8 (*Xcc*Xeg74 complexed with XG oligosaccharide), 7KMM (native *Xac*XaeA), 7KMN (native *Xac*GalD), 7KMO (*Xac*GalD complexed with galactose), 7KMP (native *Xac*Xyl31), 7KNC (*Xac*Xyl31 complexed with xylose) and 7KMQ (native *Xac*Afc95). RNA-seq data were deposited in the Gene Expression Omnibus database under accession number GSE159288. Additional data that support the findings of this study are available from the corresponding authors on reasonable request. Source data are provided with this paper.

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

## Acknowledgements

We are grateful to Dr. Felipe Rafael Torres for the discussions on the RT-qPCR analysis, to Dr. Nicolas Terrapon and the CAZy database for analyzing and establishing the novel CE20 family, to Dr. Juliana Velasco de Castro Oliveira for the support in the cultivation of *A. thaliana* and all the staff of the several facilities listed below for their technical support. We thank the Brazilian Synchrotron Light Laboratory (LNLS—CNPEM/MCTI) for the provision of time on the SAXS1 and MX2 beamlines, the Stanford Synchrotron Radiation National Accelerator Laboratory (SSRL-SLAC) for the provision of time on the BL9-2 beamline, the Brazilian Biosciences National Laboratory (LNBio—CNPEM/MCTI) for access to the crystallization (Robolab) facility, the Nuclear Magnetic Resonance (NMR) facility and the Chemistry and Natural Products Laboratory, and the Brazilian Bior-enewables National Laboratory (LNBR—CNPEM/MCTI) for the use of the Next Generation Sequencing (NGS) and Characterization of Macromolecules (MAC) open access facilities. Use of the Stanford Synchrotron Radiation Lightsource, SLAC National Accelerator Laboratory, is supported by the U.S. Department of Energy, Office of Science, Office of Basic Energy Sciences under Contract No. DE-AC02-76SF00515. The SSRL Structural Molecular Biology Program is supported by the DOE Office of Biological and Environmental Research, and by the National Institutes of Health, National Institute of General Medical Sciences (P41GM103393). The contents of this publication are solely the responsibility of the authors and do not necessarily represent the official views of NIGMS or NIH. This research was funded by São Paulo Research Foundation (FAPESP, 2015/26982-0 to M.T.M. and 2015/13684-0 to I.P.) and by Conselho Nacional de Desenvolvimento Científico e Tecnológico (CNPq, 303988-2015-5 to I.P., and 408600/2018-7 to G.F.P.). P.S.V., R.R.M., M.A.B.M., and A.G. received post-doctoral FAPESP fellowships (2016/06509-0, 2017/14253-9, 2016/19995-0 and 2019/13936-6 respectively), I.M.B. and J.A.D. received FAPESP PhD fellowships (2017/00203-0 and 2018/03724-3 respectively), and E.A.A. received CNPq PhD fellowship (158752/2015-5).

## Author contributions

P.S.V., E.A.A., E.A.L., M.R.F., R.R.M., M.A.B.M., J.B.L.C., J.A.D. performed enzyme assays. P.S.V. and L.M.Z. performed biophysical analyses. P.S.V., A.G. and M.S.B. performed carbohydrate extraction. P.S.V., E.A.A., M.A.B.M., I.P. and M.T.M. performed crystallographic studies. S.A.R. synthesized and analyzed the acetylated mono- and disaccharides. A.R.L., I.M.B., C.E.B. and P.O.G. performed gene knockout and in vivo experiments. I.M.B., D.A.A.P., G.F.P. and P.O.G. performed RNA-seq analyses. G.F.P. performed phylogenetic analyses. T.B.L., R.A.S.P., M.S.B. and F.C.G. performed and analyzed mass spectrometry experiments. I.P., P.O.G. and M.T.M. coordinated the work, analyzed the results and wrote the manuscript. P.S.V. and I.M.B. also contributed to the writing of the manuscript.

## Competing interests

The authors declare no competing interests.
