## [Peer Review File · Nature Communications]

REVIEWER COMMENTS

Reviewer #1 (Remarks to the Author):

This manuscript by Vieira et al. presents the characterisation of the xyloglucan utilisation system of the plant pathogen *Xanthomonas citri* by biochemical, structural biology, and in vivo microbiology approaches. This study is important because the many *Xanthomonas* species are phytopathogenic on a wide range of important crops, and therefore have significant effects on global food production. Likewise, xyloglucan is ubiquitous in the cell walls land plants, especially dicots, and therefore constitutes a primary target for the initial attack of these bacteria on plant, as well as providing an abundant source of carbohydrates for bacterial growth. As the authors show, this xyloglucan utilisation system is widespread and highly conserved across the *Xanthomonas* genus, the so present work unveils a previously uncharacterised and widely important nutrient acquisition system in these infamous bacteria.

The manuscript is, overall, very well written and describes a wealth of data from this multi-faceted study. The structural biology is particularly elegant, including the presentation of a new carbohydrate esterase fold. However, the following comments should be addressed to improve further the study and its presentation. Note that these comments also include requests for additional data analysis and experiments, which I expect will not be very onerous, considering the expertise of the authors.

1. p.3, paragraph 3: Please include a reference to Figure 1 at the end of this sentence, "Plant pathogens from *Xanthomonas* genus also encompass a gene cluster predicted to degrade XyGs."

2. p.4, Results, paragraph 1: Based on the existing transcriptomics data (e.g. Fig. 5), please describe in more detail the co-regulation of all of the genes in the predicted XyGUL (those shown in Fig. 1a), as well as the downstream xylose metabolism and MFS genes indicated here (cf. Fig. S22 and S23). Fig. 5 notably shows the *cirA* TBDT as the most upregulated gene during growth on xyloglucan oligosaccharides (XyGOs); how do these data compare to transcriptomics during growth on the polysaccharide? Is the companion TBDT *fhuA* upregulated to the same extent, and what are the implications of this? A more fullsome quantitative analysis of gene expression levels would help to better define the XyGUL and the associated genes. It would certainly be helpful to annotate the XyGUL genes within Fig. 5a, perhaps the downstream xylose metabolism genes, etc., as well. Also, can a composite value for the XyGUL genes be added to Fig. 5b?

2b. Connected with the comment above and especially the data in Fig. 5, whether or not the *fhuA* TBDT (encoded by XAC1768) is an integral part of the XyGUL or not is unclear. It appears that only a double-TBDT knock-out strain was produced (p. 32, top). The individual single gene knock-outs of the TBDTs should be made to determine the extent to which they might complement each other. This is especially relevant as both TBDTs are not strictly conserved among all *Xanthomonas* species (e.g. *X. cucurbitae* vs. *X. fragariae*, Fig. 1b).

3. p.4, Results, paragraph 1, last sentence; and p. 13, Discussion, paragraph 1: Based on ref. 6, it seems some *Bacteroides* XyGULs also contain GH95 fucosidases. Perhaps ref. 6 and Dejean et al. *Appl Environ Microbiol.* 2019 Oct 1;85(20):e01491-19. doi: 10.1128/AEM.01491-19 could be cited here.

4. p.4, paragraph 2: Consider replacing or augmenting refs. 12 and 13 with the following review by Vogel *Curr Opin Plant Biol.* 2008 Jun;11(3):301-7. doi: 10.1016/j.pbi.2008.03.002 (see Table 1) and the more recent ref. by Brennan et al. *Plant Physiol Biochem.* 2019 Jun;139:428-434. doi: 10.1016/j.plaphy.2019.04.005

5. p.4-p.5: Revise to "These observations indicate that this system..." and "...investigate in depth..."

6. p.5, bottom: The discussion of the aromatic residues in the *XccXeg74A* active-site ends with a rather odd conclusion. If the enzyme does not bind XyG productively, how does it hydrolyse it? I suspect the authors mean to say something else here. Perhaps a comment on the relative kinetic

efficiency of this enzyme to others in the family would be relevant in this context.

7. Throughout: Regarding enzyme nomenclature, Xeg74A, Afc95A, etc., i.e. including the "A" implies the existence of additional members of these families (e.g. "B", "C", etc.) in the *Xanthomonas* strains studied here. If there is not more than one homologue in each strain, the letter can be dropped. Genomes from related strains suggest that this may be the case, e.g.: <http://www.cazy.org/b4297.html>

8. p.6, paragraph 1: I am not convinced that the 165 glycoside hydrolase families constitute a "superfamily", since in the structural biology sense, they comprise many unrelated folds and evolutionary trajectories. Ref. 28 should also be cited here to support the statement on the current number of GH families. The last sentence in the paragraph also requires grammar revision.

9. p.7, paragraph 2: Revise to "...described in the CAZy database."

10. p. 7, last paragraph to p. 8, paragraph 1: Could the authors please add the type of signal peptidase cleavage site that is predicted for each protein to Table 1? Also, how do the authors rectify the majority prediction of the xylosidase and fucosidase at the outer membrane? The last sentence in the first paragraph on p. 8 could be construed as misleading in this regard, when compared with the data in Table 1.

11. p.8, paragraph 3: How does the sequence of the XacAfc95A compare with the previously characterised GH95 fucosidases from *Cellvibrio japonicus* (ref. 5) and *Bacteroides* species (Dejean et al. *Appl Environ Microbiol.* 2019 Oct 1;85(20):e01491-19), which the authors use as reference points for other XyGULs? I appreciate that the authors are comparing their new 3-D structure to the two existing examples in the PDB, but the *B. ovatus* alpha-L-galactosidase may not be the best representative from *Bacteroides*, due to its distinct activity. The statement on p. 9 "Additional biochemical and structural studies of more members of this yet underexplored family..." seems to overlook the data in the publications mentioned in this comment, as well as several other characterised members: http://www.cazy.org/GH95_characterized.html

12. p.8, paragraph 3: Also, throughout the manuscript the authors must rigorously use the terms "alpha-L-galactosidase" and "alpha-L-galactosidase", as well as "L-fucose" and "L-galactose" (with the "L" stereochemistry identifier), to indicate more clearly the similarity of the substrate specificity between these enzymes. It would be helpful to add in the text that L-fucose is equivalent to 6-deoxy-L-galactose.

13. p.9, last line: This abbreviated enzyme name looks off to me. Also, Table S9 should be referenced after the last sentence of this paragraph.

14. p.10, paragraph 3: See and cite together with refs. 33,36: Silipo et al. *Chemistry.* 2012 Oct 15;18(42):13395-404. doi: 10.1002/chem.201200488

15. p.11, paragraph 1; and p. 14, paragraph 1: *C. japonicus* also contains a highly specific beta-glucosidase for XyGOs, which might be inferred to indicate a theme among XyG utilisation systems, where the specialised glucosidase is not encoded within the XyGUL. Drawing this parallel would strengthen the authors' discussion. See and cite: Nelson et al. *Environ Microbiol.* 2017 Dec;19(12):5025-5039. doi: 10.1111/1462-2920.13959

16. p.11, paragraph 3: Revise to "...both types of XyGs..."

17. p.13, last paragraph of the Results section: With reference to Aspeborg et al. *BMC Evol Biol.* 2012 Sep 20;12:186. doi: 10.1186/1471-2148-12-186, please list the gene loci and indicate the subfamily membership (GH5_n) of all four GH5 members. Are these GH5 and GH9 members more likely to be "engaged in cellulose depolymerization", or mixed-linkage beta-glucan depolymerization? Is *Xanthomonas* an efficient cellulolytic organism? It is perhaps worth remembering that CMC is quite different from crystalline or semi-crystalline cellulose.

18. p. 13, Discussion, paragraph 1: While it appears from ref. 6 that *Bacteroides* do not share the

same GH35 beta-galactosidase, their corresponding XyGULs do encode this activity, from GH2. On the other hand, ref. 5 indicates that the XyGUL in *Cellvibrio japonicus*, the other species which the authors use as a reference point, does contain a GH35 alpha-galactosidase in its XyGUL. Thus, the last sentence in this paragraph glosses over these commonalities to give the reader the impression that the *Xanthomonas* XyGUL is more unique than it is - it might be considered something of a hybrid (in the conceptual sense, not in the evolutionary sense) between the *Bacteroides* and *Cellvibrio* XyGULs. See also comment 3 above. A distinguishing feature with both XyGULs, of course, is the CE. The authors should revise this paragraph accordingly.

19. p. 14, paragraph 1: Revise "As cello-oligosaccharides are common intermediate products of XyGs and cellulose depolymerization..." to "As gluco-oligosaccharides are common intermediate products of XyG and cellulose depolymerization..."

20. p. 14, paragraph 1: Following comment 15 above, it would seem that both *Xanthomonas* and *Cellvibrio* do not simply "share GH3 glucosidases" between XyGUL and cellulolytic systems, but rather have specifically evolved beta-glucosidases for XyGO hydrolysis. Perhaps this discussion could be elaborated accordingly.

21. Fig. 1b: What is the significance of the apparent deletion of a GH74 xyloglucanase in some XyGULs, which otherwise contain one or more TBDTs and exo-GHs that surround this position in the genomes? With regard to comment 17 above, is there a potential compensating xyloglucanase from another family? Perhaps this could be discussed in the text.

22. Fig. 6: Please add the corresponding gene loci for all the GHs to allow direct comparison with Fig. 1 (as is the case for the TBDTs in Fig. 6).

23. p.29-30: I understand that the authors have expertise in xyloglucan purification. Randomly acetylated mono- and disaccharides are poor analogs of the natural substrate. It would be more compelling if the authors could quantify acetate release from a native xyloglucan.

End of comments.

Reviewer #2 (Remarks to the Author):

This is a nice story regarding the xyloglucan degradation mechanism in the diverse and economically relevant *Xanthomonas* bacteria. The authors first showed that the citrus canker pathogen contains a system encompassing distinctive glycoside hydrolases, a modular carbohydrate acetyltransferase and specific membrane transporters, demonstrating that plant-associated bacteria employ distinct molecular strategies from commensal gut bacteria to cope with xyloglucans. Secondly, the authors solved the 3-D structures of the key enzymes involved in this process. Finally, the authors showed that sugars released by this system elicit the expression of several key virulence factors including the type III secretion system, a membrane-embedded apparatus to deliver effector proteins into the host cells. Generally, these findings shed light on the molecular mechanisms underpinning the intricate enzymatic machinery of *Xanthomonas* to depolymerize xyloglucans and uncover a role for this system in signaling pathways driving pathogenesis. I have the following minor concerns: (1) In figure 1b, the authors showed that all the genes in citrus canker pathogens are also present in *Xanthomonas campestris* pv. *campestris*. Why only the authors only present the homologous gene cluster in the strain Xcc 3811, not other known genomes with published genome sequences? For example, 8004, ATCC33913, B100? (2) Did the authors compare the xyloglucan degradation genes in all the citrus canker pathogens? (3) In Fig.2, why the authors only showed the 3-D structure of XccXeg74A, which is an enzyme present in *Xanthomonas campestris* pv. *campestris*, which is not causal agent of citrus canker?

Reviewer #3 (Remarks to the Author):

Summary

Vieira et. al. present a comprehensive biochemical analyses of a xyloglucan (XG) utilization locus (XyGUL from *Xanthomonas citri* pv. *Citri*, linking its degradation products (XG oligomers) to the activation of important pathogenesis pathways in the bacterium. The authors show that the XyGUL is conserved in *Xanthomonas* sp. Plant pathogens, and is involved in the regulation of virulence genes, suggesting that the XyGUL is important in *Xanthomonas* plant virulence, endowing the bacteria the ability to depolymerize xyloglucan in the plant cell walls in order to more efficiently infect the plants. The findings presented are of high value to research on carbohydrate-active enzymes, and also to the research on *Xanthomonas* virulence in important agricultural crops.

Major findings

In the detailed biochemical analyses of the enzymes of the XyGUL, the authors systematically present biochemical data supporting that the model XyGUL and accessory enzymes of *X. citri* possesses all the enzymatic activities required for the depolymerization of XG. Knockout experiments of the two XyGUL tonB-dependent receptors proved detrimental to XG growth, further supporting the function of the XyGUL.

Structural and functional experiments of the individual enzymes revealed new insights into several nifty enzymatic details; notably an unconventional substrate recognition mode in glycoside hydrolase family 74s (GH74s) conserved across *Xanthomonas* sp., a novel structural architecture for carbohydrate esterases, structural insights into the so far limited knowledge on GH95 structures, and oligomerization as a requisite for catalytic activity the *X. citri* XyGUL GH35.

The authors also present transcriptomic analyses with XG degradation products as substrate, and gene-deletion virulence assays of the XyGUL of *X. citri*. Transcriptomic analyses showed that genes important in early stage infection, and a major regulator of the type 3 secretion system, important in pathogenesis, is activated by the monomeric XG degradation products, notably galactose. However, this was also shown for other carbohydrate monomers, indicating redundancy in the signals of plant cell wall degradation. Knockout studies showed that the XyGUL is not essential for virulence in *X. citri*, indicating that the XyGUL is part of a redundant arsenal of virulence factors.

Minor comments to be addressed before publication

The work presented by Vieira et. al. is comprehensive and detailed and presents original and valuable knowledge. However, I have some minor comments that should be addressed before publication;

1. In the first paragraph of the results section, comparing the XyGUL to the PULdb, the text states: "The only common enzymatic unit between *Xanthomonas* and *Bacteroidetes* is a GH31 α -xylosidase."

This sentence is unclear, and written like this, a false statement. Searching the PULdb with for example GH74+GH31, clearly shows that several *Bacteroidetes* have PULs containing GH74 and GH31.

The authors need to rephrase this statement before publication.

2. In the final paragraph of the «XyGUL endo-enzyme exploits arginine-carbohydrate interactions» section of the results, the text reads:

"It is prominent the systematic replacement of aromatic residues by arginine in *Xanthomonas* GH74 enzymes, an unconventional strategy for substrate recognition in the glycoside hydrolase superfamily, which encompasses over 165 families with dozens of activities predominantly relying on aromatic CH-(π) interactions for carbohydrate binding."

I believe the authors want to convey that this strategy is seen only in *Xanthomonas* sp GH74s,

unlike in all other GH families. However, this sentence is unclear and needs rephrasing before publication.

3. In the section on the "Xanthomonas intracellular cascade for XyG oligosaccharide breakdown", on the GH95T375H mutation, the text reads:

"This finding points out to a more elaborated mechanism of substrate selectivity in this family."

The text should be changed to "... points to a more elaborate mechanism".

4. In the same section, in the final paragraph on beta-glucosidases, the text reads:

"These three β -glucosidases are expressed in the presence of XyGOs, ...".

Where is this shown in the experimental data? These should be listed in supplementary table 10, and this should be referenced in the text.

5. In the discussion section, the paragraph starting with "From a mechanistic point of view", the final sentence reads:

"In addition, it is prominent the role of ancillary domains and oligomerization in the function of Xanthomonas XyGUL enzymes (Fig. 4)."

This should be changed for better grammar, for example: " In addition, the role of ancillary domains and oligomerization is prominent in the function of Xanthomonas XyGUL enzymes (Fig. 4)."

REVIEWER COMMENTS

Reviewer #1 (Remarks to the Author):

This manuscript by Vieira et al. presents the characterisation of the xyloglucan utilisation system of the plant pathogen *Xanthomonas citri* by biochemical, structural biology, and in vivo microbiology approaches. This study is important because the many *Xanthomonas* species are phytopathogenic on a wide range of important crops, and therefore have significant effects on global food production. Likewise, xyloglucan is ubiquitous in the cell walls land plants, especially dicots, and therefore constitutes a primary target for the initial attack of these bacteria on plant, as well as providing an abundant source of carbohydrates for bacterial growth. As the authors show, this xyloglucan utilisation system is widespread and highly conserved across the *Xanthomonas* genus, the so present work unveils a previously uncharacterised and widely important nutrient acquisition system in these infamous bacteria.

The manuscript is, overall, very well written and describes a wealth of data from this multi-faceted study. The structural biology is particularly elegant, including the presentation of a new carbohydrate esterase fold. However, the following comments should be addressed to improve further the study and its presentation. Note that these comments also include requests for additional data analysis and experiments, which I expect will not be very onerous, considering the expertise of the authors.

1. p.3, paragraph 3: Please include a reference to Figure 1 at the end of this sentence, "Plant pathogens from *Xanthomonas* genus also encompass a gene cluster predicted to degrade XyGs."

A: The figure was mentioned at the end of this sentence.

2. p.4, Results, paragraph 1: Based on the existing transcriptomics data (e.g. Fig. 5), please describe in more detail the co-regulation of all of the genes in the predicted XyGUL (those shown in Fig. 1a), as well as the downstream xylose metabolism and MFS genes indicated here (cf. Fig. S22 and S23). Fig. 5 notably shows the *cirA* TBDT as the most upregulated gene during growth on xyloglucan oligosaccharides (XyGOs); how do these data compare to transcriptomics during growth on the polysaccharide? Is the companion TBDT *fhuA* upregulated to the same extent, and what are the implications of this? A more fullsome quantitative analysis of gene expression levels would help to better define the XyGUL and the associated genes. It would certainly be helpful to annotate the XyGUL genes within Fig. 5a, perhaps the downstream xylose metabolism genes, etc., as well. Also, can a composite value for the XyGUL genes be added to Fig. 5b?

A: We described in more detail the expression pattern of the genes in the predicted XyGUL and the downstream genes related to xylose metabolism and inner membrane transporter (MFS) including a new figure 5b describing the expression fold change of such genes in the presence of XyGOs in relation to glucose. From this new figure, it is clear the increased expression levels of XyGUL and downstream genes, but not to the same extent as *cirA* TBDT gene. Transcriptional data using the polysaccharide as the main carbon source would probably show a more pronounced increase in the expression levels of the CAZyme-encoding XyGUL genes, but *Xanthomonas* growth in such condition was lower than the scale required for RNA extraction, hampering such transcriptomic assays. It would be expected since under typical *in planta*

conditions *Xac* can utilize a number of cell wall polysaccharides as nutrient source to support growth and a successful colonization. This observation is also in agreement with our findings that indicate other roles to this PUL than nutrition. Therefore, other characteristics were also used to define the XyGUL boundaries and associated genes, such as gene co-directionality, conservation of the predicted XyGUL and associated genes in other *Xanthomonas* species, the functional characterization of CAZymes encoded by this gene cluster, and the effect of TBDTs knockouts on XyGOs utilization, as described below.

The companion TBDT *fhuA* was not upregulated as *cirA* was in our assays, indicating that *cirA* might have a more relevant role in the uptake of XyGOs than *fhuA*. To address this issue, we generated strains containing the single gene knockout of *cirA* or *fhuA*, which revealed that *cirA* is required to support the full uptake of XyGOs in *Xac*. However, the growth of *Xac* in XyGOs was more severely impaired by the double TBDT knockout than by the single *cirA* knockout, supporting that *fhuA* might yet have a minor role in XyGOs uptake. We clarified this issue in the text (p. 8 paragraph 1) and reproduced here.

“Supporting this hypothesis, the knockout of both TonB-dependent transporters (TBDTs) encoded by the XyGUL (XAC1768 and XAC1769) was highly detrimental to *X. citri* growth with XyGOs as carbon source, but not to the growth with a mixture of its monosaccharides, indicating that depolymerization of XyGOs occurs after passing the outer-membrane using specific TBDTs. Individual knockout of these transporters supports a major role for XAC1769 in XyGOs uptake, being sufficient to maintain the transport in the absence of XAC1768 (Supplementary Fig. 11).”

2b. Connected with the comment above and especially the data in Fig. 5, whether or not the *fhuA* TBDT (encoded by XAC1768) is an integral part of the XyGUL or not is unclear. It appears that only a double-TBDT knock-out strain was produced (p. 32, top). The individual single gene knock-outs of the TBDTs should be made to determine the extent to which they might complement each other. This is especially relevant as both TBDTs are not strictly conserved among all *Xanthomonas* species (e.g. *X. cucurbitae* vs. *X. fragariae*, Fig. 1b).

A. We made the individual single gene knockouts of XAC1768 (*fhuA*) and XAC1769 (*cirA*) and their growth profiles in XyGOs support that XAC1769 plays a major role in XyGOs uptake, corroborating transcriptomic analysis. Please also see the answer to question 2a for more details.

3. p.4, Results, paragraph 1, last sentence; and p. 13, Discussion, paragraph 1: Based on ref. 6, it seems some *Bacteroides* XyGULs also contain GH95 fucosidases. Perhaps ref. 6 and Dejean et al. *Appl Environ Microbiol.* 2019 Oct 1;85(20):e01491-19. doi: 10.1128/AEM.01491-19 could be cited here.

A: We agree with the reviewer and reformulated these paragraphs to better describe the similarities and dissimilarities between *Xanthomonas* and *Bacteroides* XyGULs. We also included the reference to the study of Dejean and coworkers as an example in the results section (pag. 4, paragraph 1). Please see below the new sentence:

“A search in the polysaccharide-utilization loci database (PULDB) did not result in any similar organization in *Bacteroidetes*, except for the clustering of three or two XyG-related genes in some species. In characterized *Bacteroides* XyGULs, the clustering of GH31 and GH95 genes has

been observed, but in association with other carbohydrate-active enzymes (CAZymes). The synteny of GH31, GH35 and GH95 genes was reported in the XyGUL from the saprophyte *Celvibrio japonicus*, but not physically linked to endoxyloglucanases or esterases genes."

4. p.4, paragraph 2: Consider replacing or augmenting refs. 12 and 13 with the following review by Vogel Curr Opin Plant Biol. 2008 Jun;11(3):301-7. doi: 10.1016/j.pbi.2008.03.002 (see Table 1) and the more recent ref. by Brennan et al. Plant Physiol Biochem. 2019 Jun;139:428-434. doi: 10.1016/j.plaphy.2019.04.005

A: The recommended references were added.

5. p.4-p.5: Revise to "These observations indicate that this system..." and "...investigate in depth..."

A: The sentence was revised accordingly to the suggestion.

6. p.5, bottom: The discussion of the aromatic residues in the XccXeg74A active-site ends with a rather odd conclusion. If the enzyme does not bind XyG productively, how does it hydrolyse it? I suspect the authors mean to say something else here. Perhaps a comment on the relative kinetic efficiency of this enzyme to others in the family would be relevant in this context.

A: The sentence was revised to better express that the lack of aromatic residues in some subsites of XccXeg74A might be compensated by other mechanisms of substrate recognition, which are described in the following paragraph of the manuscript. The revised sentence reads "It implicates that *Xanthomonas* GH74 enzymes might have evolved molecular strategies complementary to the conventional stacking mechanism for carbohydrate binding."

7. Throughout: Regarding enzyme nomenclature, Xeg74A, Afc95A, etc., i.e. including the "A" implies the existence of addition members of these families (e.g. "B", "C", etc.) in the *Xanthomonas* strains studied here. If there is not more than one homologue in each strain, the letter can be dropped. Genomes from related strains suggest that this may be the case, e.g.: <http://www.cazy.org/b4297.html>

A: As requested, we changed the nomenclature for XccXeg74A (*XccXeg74*), XccXyl31A (*XccXyl31*) and XccAfc95A (*XccAfc95*). According to the genome annotation for *Xanthomonas citri* pv. *citri* 306 (<http://www.cazy.org/b94.html>), it contains four enzymes from family GH35, thus we maintained the nomenclature for XAC1772 (*XccGalD*). We also noticed the presence of a paralogue enzyme of XccXaeA in the genome (XAC4228) and also maintained the proposed nomenclature.

8. p.6, paragraph 1: I am not convinced that the 165 glycoside hydrolase families constitute a "superfamily", since in the structural biology sense, they comprise many unrelated folds and evolutionary trajectories. Ref. 28 should also be cited here to support the statement on the current number of GH families. The last sentence in the paragraph also requires grammar revision.

A: We revised the sentence accordingly and added the indicated reference. Now it reads “This molecular strategy of carbohydrate recognition based on arginine residues observed in *Xanthomonas* GH74 enzymes is a distinguishing feature among all GH families, which typically rely on aromatic CH- π interactions for carbohydrate binding.”

9. p.7, paragraph 2: Revise to "...described in the CAZy database."

A: The sentence was revised as suggested.

10. p. 7, last paragraph to p. 8, paragraph 1: Could the authors please add the type of signal peptidase cleavage site that is predicted for each protein to Table 1? Also, how do the authors rectify the majority prediction of the xylosidase and fucosidase at the outer membrane? The last sentence in the first paragraph on p. 8 could be construed as misleading in this regard, when compared with the data in Table 1.

A: As requested, we added the type of signal peptidase cleavage site in Supplementary Table 1, which shed light on this issue revealing that the β -Galactosidase, the β -glucosidase XAC1793 and the endoglucanase XAC2522 are lipoproteins, likely anchored in the outer membrane, according to the type of amino acid at +2 position after the SPII cleavage site. Concerning the xylosidase and fucosidase, we complemented the analysis by verifying if they display N- or C-terminal transmembrane helices for membrane anchoring. Since they neither have a SPII cleavage site nor a predicted transmembrane helix, we considered more parsimonious to suggest their location at the periplasm, as predicted by the CELLO program. We rewrote the paragraph to better describe our conclusions based on the predictions shown in the Supplementary Table 1. Note that the supplementary table 1 was revised to clarify the multiple criteria we used to propose the probable subcellular localization of each enzyme.

11. p.8, paragraph 3: How does the sequence of the XacAfc95A compare with the previously characterised GH95 fucosidases from *Cellvibrio japonicus* (ref. 5) and *Bacteroides* species (Dejean et al. Appl Environ Microbiol. 2019 Oct 1;85(20):e01491-19), which the authors use as reference points for other XyGULs? I appreciate that the authors are comparing their new 3-D structure to the two existing examples in the PDB, but the *B. ovatus* alpha-L-galactosidase may not be the best representative from *Bacteroides*, due to its distinct activity. The statement on p. 9 "Additional biochemical and structural studies of more members of this yet underexplored family..." seems to overlook the data in the publications mentioned in this comment, as well as several other characterised members: http://www.cazy.org/GH95_characterized.html

A: XacAfc95A shares 38-40% sequence identity with the characterised GH95 fucosidase from *C. japonicus* (CjAfc95A) and *Bacteroides* species (Dejean et al., 2019). We included this information in the text. We agree with the reviewer and extended our sequence comparisons with these and other GH95 characterised enzymes listed in the CAZY database (new supplementary figure 16). We deleted the statement on p. 9 and revised the paragraph to explicit our findings based on this broader comparison with all characterised members. Please see below.

“Based on structural analyses, they proposed that the presence of a threonine at this position would allow a hydrogen bond with the L-galactose O6 atom, whereas a histidine would contribute to aliphatic interactions with the L-fucose C6 methyl group. However, XacAfc95, which shares nearly 40% sequence identity with characterised GH95 α -1,2-L-fucosidases

involved in XyG depolymerization, contains a threonine at the referred position and showed high specificity to L-fucose, contraposing the initial role proposed for this residue as a determinant for L-galactose preference (Fig 4b, Supplementary Tables 5,6,7, Supplementary Fig. 12h). In addition, the mutation T395H did not result in any change of substrate preference, supporting a less relevant role of this polymorphic position to selectivity in the GH95 family (Supplementary Fig. 15a). Besides *XacAfc95*, another characterized GH95 α -1,2-L-fucosidase (Blon_2335) conserves a threonine at this polymorphic position, corroborating this hypothesis (Supplementary Fig. 16). These findings point to a more elaborate mechanism of substrate selectivity in the GH95 family that is not limited to direct interactions with the residues forming the -1 subsite.”

12. p.8, paragraph 3: Also, throughout the manuscript the authors must rigorously use the terms "alpha-L-galactosidase" and "alpha-L-galactosidase", as well as "L-fucose" and "L-galactose" (with the "L" stereochemistry identifier), to indicate more clearly the similarity of the substrate specificity between these enzymes. It would be helpful to add in the text that L-fucose is equivalent to 6-deoxy-L-galactose.

A: We added the identifier throughout the text and included the information that L-fucose is equivalent to 6-deoxy-L-galactose.

13. p.9, last line: This abbreviated enzyme name looks off to me. Also, Table S9 should be referenced after the last sentence of this paragraph.

A: We changed the abbreviation to *CjGH35A*, as it is mentioned in the original paper. We also added the reference to Table S10 (older Table S9) in the sentence.

14. p.10, paragraph 3: See and cite together with refs. 33,36: Silipo et al. Chemistry. 2012 Oct 15;18(42):13395-404. doi: 10.1002/chem.201200488

A: Thank you for the relevant literature. As suggested, we added the reference.

15. p.11, paragraph 1; and p. 14, paragraph 1: *C. japonicus* also contains a highly specific beta-glucosidase for XyGOs, which might be inferred to indicate a theme among XyG utilisation systems, where the specialised glucosidase is not encoded within the XyGUL. Drawing this parallel would strengthen the authors' discussion. See and cite: Nelson et al. Environ Microbiol. 2017 Dec;19(12):5025-5039. doi: 10.1111/1462-2920.13959

A: We thank the reviewer for the nice comment and inspired by the work of Nelson et al. we performed additional experiments to address this issue. The three *X. citri* β -glucosidases were assayed against GXXG obtained by the action of *XacXyl31* on XXXG. Substrates and products were monitored by mass spectrometry. All β -glucosidases were active on GXXG, but with distinct rates. We included these new results and made a parallel with the work of Nelson et al, 2017 in the main text. Please see below.

“These three β -glucosidases were expressed in the presence of XyGOs with higher expression levels of *XacBgl3C* and *XacBgl3B* compared to *XacBgl3A* (Supplementary Table 11). All three enzymes were capable to release the non-reducing glucosyl moiety from the XyG-

derived oligosaccharides GXXG and GXG, which are the products of *XacXyl31* using XXXG and XXG as substrates, respectively (Supplementary Fig. 21,22).

XacBgl3B displayed the highest activity on GXXG and GXG substrates and is predicted to be an outer-membrane associated protein, probably exposed to the periplasm (Supplementary Table 1). *XacBgl3A*, although predicted to be periplasmic, seems to be more specific to cleave β -1,3-glucooligosaccharides instead of XyGOs-derived β -1,4-glucooligosaccharides (Supplementary Fig. 23,24, Supplementary Tables 1,7). *XacBgl3C* displayed a more generalist substrate profile and is predicted to be cytoplasmic, indicating that it might support the final steps of glucooligosaccharides cleavage coming from different sources (Supplementary Fig. 23,24, Supplementary Tables 1,7).

Together, these findings point to *XacBgl3B* as being the major β -glucosidase to alternate with *XacXyl31* on the breakdown of XyGOs intermediates until reaching the final substrate β -1,4-glucobiose. At this last step, *XacBgl3B* seems also to play an important role, since it was the most efficient on β -1,4-glucobiose cleavage, compared to *XacBgl3A* and *XacBgl3C* (Supplementary Fig. 23,24, Supplementary Table 7)."

16. p.11, paragraph 3: Revise to "...both types of XyGs..."

A: We changed the sentence as suggested.

17. p.13, last paragraph of the Results section: With reference to Aspeborg et al. BMC Evol Biol. 2012 Sep 20;12:186. doi: 10.1186/1471-2148-12-186, please list the gene loci and indicate the subfamily membership (GH5_n) of all four GH5 members. Are these GH5 and GH9 members more likely to be "engaged in cellulose depolymerization", or mixed-linkage beta-glucan depolymerization? Is *Xanthomonas* an efficient cellulolytic organism? It is perhaps worth remembering that CMC is quite different from crystalline or semi-crystalline cellulose.

A: As suggested, the four subfamilies of the GH5 family were included following Aspeborg, et al. 2012. *Xac* is not an efficient cellulolytic organism, in the sense that it lacks cellulosomes, or LPMOs or cellobiohydrolases (GH6). In particular, the presence of GH6 CBHs has been correlated with high cellulolytic activity of some *Xanthomonas* species and seems to act as a phenotypic switch between vascular and nonvascular plant pathogenesis (Gluck-Taler et al., 2020). About the potential polysaccharide target for these GH5 and GH9, the initial activity screening points to a more likely role on other β -glucans than to a specific function on cellulose depolymerization. We intend to address in details the substrate specificity of the several *Xac* GH5 and GH9 endoglucanases in another study, but we agree that the last sentence of p.13 is misleading in this regard. So, we removed this sentence from the text and focused our discussion on the results of the activity assays on XyG for these endoglucanases, which is instrumental to this work (please see below).

"A search for endo- β -1,4-glucanases in *X. citri* genome resulted in one GH9 member (XAC2522, *XacEgl9*), a GH8 member (XAC3516, *XacCel846*), and five putative GH5 glucanases from subfamilies GH5_1 (XAC0612, *XacEngXCA*) and GH5_5 (XAC0028, *XacEgl5A*, XAC0029, *XacEgl5B* and XAC0030, *XacEgl5C*) added by XAC0346 not yet assigned to a subfamily. The recombinant

production and activity assays of these enzymes revealed that only XacEgl9, XacEgl5B and XacEngXCA are able to cleave XyG, being XacEgl9 the most active (Supplementary Table 16). This result indicates that these glucanases might compensate the absence of the GH74 xyloglucanase in the Δ XAC1770 mutant. Characterization of XacEgl9 revealed kinetics parameters on XyG akin to those found for XacXeg74 (Supplementary Table 7, Supplementary Fig. 5g,12i), supporting the putative role of other endo- β -1,4-glucanases for XyG cleavage in strains lacking the GH74 enzyme."

18. p. 13, Discussion, paragraph 1: While it appears from ref. 6 that Bacteroides do not share the same GH35 beta-galactosidase, their corresponding XyGULs do encode this activity, from GH2. On the other hand, ref. 5 indicates that the XyGUL in Cellvibrio japonicus, the other species which the authors use as a reference point, does contain a GH35 alpha-galactosidase in its XyGUL. Thus, the last sentence in this paragraph glosses over these commonalities to give the reader the impression that the Xanthomonas XyGUL is more unique than it is - it might be considered something of a hybrid (in the conceptual sense, not in the evolutionary sense) between the Bacteroides and Cellvibrio XyGULs. See also comment 3 above. A distinguishing feature with both XyGULs, of course, is the CE. The authors should revise this paragraph accordingly.

A: We agree with the reviewer and it was revised accordingly. The last sentence was completely rewritten, highlighting as the main distinguishing feature the new CE family, as suggested. Please, also see the answer for comment 3 above that is related to this topic.

19. p. 14, paragraph 1: Revise "As cello-oligosaccharides are common intermediate products of XyGs and cellulose depolymerization..." to "As gluco-oligosaccharides are common intermediate products of XyG and cellulose depolymerization..."

A: The text was revised as properly indicated.

20. p. 14, paragraph 1: Following comment 15 above, it would seem that both Xanthomonas and Cellvibrio do not simply "share GH3 glucosidases" between XyGUL and cellulolytic systems, but rather have specifically evolved beta-glucosidases for XyGO hydrolysis. Perhaps this discussion could be elaborated accordingly.

A. We agree with the reviewer and this sentence was written. For more details on this issue, please see the answer to comment 15.

21. Fig. 1b: What is the significance of the apparent deletion of a GH74 xyloglucanase in some XyGULs, which otherwise contain one or more TBDTs and exo-GHs that surround this position in the genomes? With regard to comment 17 above, is there a potential compensating xyloglucanase from another family? Perhaps this could be discussed in the text.

A: Regarding the lack of GH74 in the XyGUL of some species, we found in their respective genomes the presence of other endoglucanases that might compensate its absence, such as a full-length GH12 member (which in Xac is truncated) and orthologs of XacEgl9 and XacEgl5B, which display xyloglucanase activity according to our results (Supplementary figure 2 and supplementary table 16). Moreover, as showed by the Xac GH74 knockout, other

endoglucanases encoded outside the XyGUL are sufficient to maintain a xyloglucanase activity akin to the WT in plate assays, supporting a redundancy in *Xanthomonas* bacteria to this central activity for XyG degradation. We included a sentence suggesting this functional compensation mediated by other endoglucanases in p. 4 (Results section, paragraph 2) and in p. 14 (last results subsection, last paragraph).

p. 4 (Results section, paragraph 2) - “The XyGUL is highly conserved across the *Xanthomonas* genus regardless the broad range of hosts (monocots and dicots) and tissue specificity (mesophyll or xylem vessels) (Fig. 1b). Few of them have lost the predicted GH74 xyloglucanase, but endoglucanases encoded outside the XyGUL, such as GH5, GH9 and GH12 members, may compensate its absence. The only exceptions that lack most of XyGUL genes are *Xanthomonas* species colonizing gramineous monocots such as *X. oryzae* (rice), *X. translucens* (wheat) and *X. albilineans* (sugarcane). This apparent loss of XyG-degrading capacity correlates with the typically lower contents of XyG in the cell walls of these plants.”

p. 14 (last results subsection, last paragraph) - “A search for endo- β -1,4-glucanases in *X. citri* genome resulted in one GH9 member (XAC2522, *XacEgl9*), a GH8 member (XAC3516, *XacCel8*), and five putative GH5 glucanases from subfamilies GH5_1 (XAC0612, *XacEngXCA*) and GH5_5 (XAC0028, *XacEgl5A*, XAC0029, *XacEgl5B* and XAC0030, *XacEgl5C*) added by XAC0346 not yet assigned to a subfamily. The recombinant production and activity assays of these enzymes revealed that only *XacEgl9*, *XacEgl5B* and *XacEngXCA* are able to cleave XyG, being *XacEgl9* the most active (Supplementary Table 16). This result indicates that these glucanases might compensate the absence of the GH74 xyloglucanase in the Δ XAC1770 mutant. Characterization of *XacEgl9* revealed kinetics parameters on XyG akin to those found for *XacXeg74* (Supplementary Table 7, Supplementary Fig. 5g,12i), supporting the putative role of other endo- β -1,4-glucanases for XyG cleavage in strains lacking the GH74 enzyme.”

22. Fig. 6: Please add the corresponding gene loci for all the GHs to allow direct comparison with Fig. 1 (as is the case for the TBDTs in Fig. 6).

A. We added the corresponding gene loci for all GHs and TBDTs in Fig. 6.

23. p.29-30: I understand that the authors have expertise in xyloglucan purification. Randomly acetylated mono- and disaccharides are poor analogs of the natural substrate. It would be more compelling if the authors could quantify acetate release from a native xyloglucan.

A. Thank you for this suggestion. It was very insightful considering the appeal of this result. Based on a protocol described in the literature (Pauly et al., *The Plant Journal*, 20: 629-639. <https://doi.org/10.1046/j.1365-313X.1999.00630.x>), we were able to extract analytical amounts of acetylated xyloglucan oligosaccharides from *Arabidopsis thaliana* cell wall using *XacXeg74*. Later, we monitored *XacXaeA* activity by ESI mass spectrometry, as illustrated by the new Supplementary Figure 7 and described by the Methodology. As a XyGUL member, the enzyme showed activity on the acetylated oligosaccharides derived from xyloglucan, independent of the branching types. After 2 hrs incubation, almost all detectable acetylated oligosaccharide peaks were expressively reduced and totally disappeared after 24 h, demonstrating the role of *XacXaeA* in the deacetylation of native substrates.

Reviewer #2 (Remarks to the Author):

This is a nice story regarding the xyloglucan degradation mechanism in the diverse and economically relevant *Xanthomonas* bacteria. The authors first showed that the citrus canker pathogen contains a system encompassing distinctive glycoside hydrolases, a modular carbohydrate acetyltransferase and specific membrane transporters, demonstrating that plant-associated bacteria employ distinct molecular strategies from commensal gut bacteria to cope with xyloglucans. Secondly, the authors solved the 3-D structures of the key enzymes involved in this process. Finally, the authors showed that sugars released by this system elicit the expression of several key virulence factors including the type III secretion system, a membrane-embedded apparatus to deliver effector proteins into the host cells. Generally, these findings shed light on the molecular mechanisms underpinning the intricate enzymatic machinery of *Xanthomonas* to depolymerize xyloglucans and uncover a role for this system in signaling pathways driving pathogenesis. I have the following minor concerns:

1) In figure 1b, the authors showed that all the genes in citrus canker pathogens are also present in *Xanthomonas campestris* pv. *campestris*. Why only the authors only present the homologous gene cluster in the strain Xcc 3811, not other known genomes with published genome sequences? For example, 8004, ATCC33913, B100 ?

A. In figure 1b, we included only one representative genome for each strain, prioritizing the illustration of *Xanthomonas* species and pathovars diversity. We verified that other *X. campestris* pv. *campestris* genomes (8004, ATCC33913, B100) also conserve the genes predicted in the Xac XyGUL (except for ATCC33913 strain which displays a truncated version of the *cirA* TBDT – see Fig. R1), indicating that Xcc 3811 can be used as a *Xanthomonas campestris* pv. *campestris* representative.

Figure R1. XyGUL conservation in *Xanthomonas campestris* strains. The XyGUL conservation was evaluated in all complete genomes of *X. campestris* pv. *campestris* currently available at Refseq database.

(2) Did the authors compare the xyloglucan degradation genes in all the citrus canker pathogens?

A: To address this issue, we evaluated XyGUL conservation in all complete genomes of *X. citri* pv. *citri* (33) currently available at Refseq database. The analysis revealed that except for *X. citri* pv. *citri* strain LH201 that has a truncated version of GH74 xyloglucanase and *X. citri* pv. *citri* strain TX160149 containing truncated versions of both TonB dependent receptor, respectively, all the other 31 genomes analyzed present the same genomic organization. We included this analysis in the supplementary material and cited it in the first paragraph of the results section (supplementary figure 1). It is worth to note that only *X. citri* can cause citrus canker.

(3) In Fig.2, why the authors only showed the 3-D structure of XccXeg74A, which is an enzyme present in *Xanthomonas campestris* pv. *campestris*, which is not causal agent of citrus canker?

A: We used XccXeg74 for the structural studies because we were unable to obtain crystals of XacXeg74. Thus, considering the Campbell principle, we tried a very close orthologue that would be representative of *Xanthomonas* GH74 xyloglucanase. XccXeg74A and XacXeg74 shares 84% sequence identity (89% sequence similarity) and all the functionally relevant regions discussed in the text are fully conserved in both enzymes (please see supplementary fig. 4).

Reviewer #3 (Remarks to the Author):

Summary

Vieira et. al. present a comprehensive biochemical analyses of a xyloglucan (XG) utilization locus (XyGUL from *Xanthomonas citri* pv. *Citri*, linking its degradation products (XG oligomers) to the activation of important pathogenesis pathways in the bacterium. The authors show that the XyGUL is conserved in *Xanthomonas* sp. Plant pathogens, and is involved in the regulation of virulence genes, suggesting that the XyGUL is important in *Xanthomonas* plant virulence, endowing the bacteria the ability to depolymerize xyloglucan in the plant cell walls in order to more efficiently infect the plants. The findings presented are of high value to research on carbohydrate-active enzymes, and also to the research on *Xanthomonas* virulence in important agricultural crops.

Major findings

In the detailed biochemical analyses of the enzymes of the XyGUL, the authors systematically present biochemical data supporting that the model XyGUL and accessory enzymes of *X. citri* possesses all the enzymatic activities required for the depolymerization of XG. Knockout experiments of the two XyGUL tonB-dependent receptors proved detrimental to XG growth, further supporting the function of the XyGUL.

Structural and functional experiments of the individual enzymes revealed new insights into

several nifty enzymatic details; notably an unconventional substrate recognition mode in glycoside hydrolase family 74s (GH74s) conserved across *Xanthomonas* sp., a novel structural architecture for carbohydrate esterases, structural insights into the so far limited knowledge on GH95 structures, and oligomerization as a requisite for catalytic activity the *X. citri* XyGUL GH35.

The authors also present transcriptomic analyses with XG degradation products as substrate, and gene-deletion virulence assays of the XyGUL of *X. citri*. Transcriptomic analyses showed that genes important in early stage infection, and a major regulator of the type 3 secretion system, important in pathogenesis, is activated by the monomeric XG degradation products, notably galactose. However, this was also shown for other carbohydrate monomers, indicating redundancy in the signals of plant cell wall degradation. Knockout studies showed that the XyGUL is not essential for virulence in *X. citri*, indicating that the XyGUL is part of a redundant arsenal of virulence factors.

Minor comments to be addressed before publication

The work presented by Vieira et. al. is comprehensive and detailed and presents original and valuable knowledge. However, I have some minor comments that should be addressed before publication;

1. In the first paragraph of the results section, comparing the XyGUL to the PULdb, the text states: "The only common enzymatic unit between *Xanthomonas* and *Bacteroidetes* is a GH31 α -xylosidase." This sentence is unclear, and written like this, a false statement. Searching the PULdb with for example GH74+GH31, clearly shows that several *Bacteroidetes* have PULs containing GH74 and GH31. The authors need to rephrase this statement before publication.

A: We agree with the reviewer and this statement was rephrased to better describe the similarities and dissimilarities between *Xanthomonas* and *Bacteroides* XyGULs as also pointed by the reviewer #1 (p. 4, paragraph 1). Please also see the answer to comment #3 from reviewer #1 that also addresses this point.

2. In the final paragraph of the «XyGUL endo-enzyme exploits arginine-carbohydrate interactions» section of the results, the text reads: "It is prominent the systematic replacement of aromatic residues by arginine in *Xanthomonas* GH74 enzymes, an unconventional strategy for substrate recognition in the glycoside hydrolase superfamily, which encompasses over 165 families with dozens of activities predominantly relying on aromatic CH-(π) interactions for carbohydrate binding." I believe the authors want to convey that this strategy is seen only in *Xanthomonas* sp GH74s, unlike in all other GH families. However, this sentence is unclear and needs rephrasing before publication.

A: As suggested, we reformulated the sentence. Now it reads "This molecular strategy of carbohydrate recognition based on arginine residues observed in *Xanthomonas* GH74 enzymes is a distinguishing feature among all GH families, which typically rely on aromatic CH- π interactions for carbohydrate binding."

3. In the section on the "Xanthomonas intracellular cascade for XyG oligosaccharide breakdown", on the GH95T395H mutation, the text reads: "This finding points out to a more elaborated mechanism of substrate selectivity in this family." The text should be changed to "... points to a more elaborate mechanism".

A: We changed the sentence in accordance to the suggestion.

4. In the same section, in the final paragraph on beta-glucosidases, the text reads: "These three β -glucosidases are expressed in the presence of XyGOs, ...". Where is this shown in the experimental data? These should be listed in supplementary table 10, and this should be referenced in the text.

A: We included this information in the supplementary table 11 and referenced it in the text.

-5. In the discussion section, the paragraph starting with "From a mechanistic point of view", the final sentence reads: "In addition, it is prominent the role of ancillary domains and oligomerization in the function of Xanthomonas XyGUL enzymes (Fig. 4)." This should be changed for better grammar, for example: " In addition, the role of ancillary domains and oligomerization is prominent in the function of Xanthomonas XyGUL enzymes (Fig. 4)."

A: The text was edited as suggested.

REVIEWERS' COMMENTS

Reviewer #1 (Remarks to the Author):

In this revision, the authors have carefully addressed all of my previous comments. The inclusion of additional, non-trivial, experimental data is especially appreciated, i.e. single TBDT gene knock-out analysis and demonstration of esterase activity on XyGOs extracted from a native plant source.

Based upon this new version, I have several minor additional comments:

1. The revised text including additional references is much more clearly presented. However, there are a number of grammatical issues throughout the manuscript, which should be carefully read again and corrected. Some examples are:

- line 91: "indicating to be a conserved"
- line 137: "Many of XyGUL-carrying Xanthomonas"
- line 227: "the CAZy"
- line 264: "Id 42.88"
- line 329: "are not engaged to form oligomeric interfaces in none of the GH31 members"
- line 340: "non-reducing"
- lines 406 and 423: "rely" should be "relies"
- lines 416 and 445: "added by" is used incorrectly.
- line 447: "being XacEgl9 the most active"
- line 456: "infect" should be "infects" (genus is singular)
- line 507: "Similarly, to that" (no comma needed)
- line 518: "unveils at mechanistic level"

2. Line 166-174: I find this section logically a bit confusing: Isn't a lack of aromatic residues in the +3 and +5 subsites exactly what one would expect for an endo-dissociative enzyme, which XccXeg74 is? I don't think there is any need to invoke "molecular strategies complementary to the conventional stacking mechanism for carbohydrate binding". Certainly, the residues that are there may hydrogen-bond to the substrate, but this is - simply speaking - not enough to retain the substrate for endo-processivity. I would suggest replacing the sentence on lines 172-174 with something like "The lack of aromatic residues in subsites +3 and +5 of XccXeg74 is, therefore, consistent with its endo-dissociative mode of action on XyG."

3. Lines 189-237: As the novel CE is indicated to be a periplasmic enzyme, this section should come after the paragraph ending on line 250. In particular, the author show that the esterase is active on the primary hydrolysis products of the GH74 enzyme, so it can be inferred that the CE does not require any of the GHs, which are described in Line 251 and down, to act first.

4. Can the authors please confirm that they have been in contact with the curators of the CAZy Database and that a new CE family has, or will be, established?

5. Lines 216 and 233: What is a "beta-Gal domain"? I don't see this described specifically, and it is unclear whether the authors mean that this is a GH, CBM, or other structural domain. Obviously, with a name like "beta-Gal", any predicted carbohydrate activity would be relevant to discuss in light of the XyG structure.

6. Line 207-209: This sentence is confusing, because the authors show explicitly that the esterase indeed has activity on oligosaccharides. Please revise to clarify the intention of this text.

7. Line 248-250: Please explicitly describe in a following sentence what was observed for the XAC1769 knock-out.

8. Line 350-352: This discussion of transcriptomic data comes before the major section on this (lines 373-376 and beyond). Perhaps some reference to "see section below" would be valuable to inform the reader that the glucosidase transcriptomic data is part of a larger dataset to follow.

9. Line 393 and/or line 469: Please include an explanation of the sources of these two types of XyG, with appropriate literature reference(s).

10. Methods, "Protein crystallization, X-ray data collection and structure determination" section: Please use Privateer to validate all carbohydrate complexes and provide the output as a supplementary table (complementary to Supplementary Table 4). See and cite Nat Struct Mol Biol. 2015 Nov;22(11):833-4. doi: 10.1038/nsmb.3115

End of comments.

Reviewer #1 (Remarks to the Author):

In this revision, the authors have carefully addressed all of my previous comments. The inclusion of additional, non-trivial, experimental data is especially appreciated, i.e. single TBDT gene knock-out analysis and demonstration of esterase activity on XyGOs extracted from a native plant source.

A: We would like to thank you once more for the previous insights that significantly contributed to improve our work. We carefully analyzed the new comments and changed accordingly. Please, find below a point-by-point response to the additional comments.

Based upon this new version, I have several minor additional comments:

1. The revised text including additional references is much more clearly presented. However, there are a number of grammatical issues throughout the manuscript, which should be carefully read again and corrected. Some examples are:

- line 91: "indicating to be a conserved"
- line 137: "Many of XyGUL-carrying Xanthomonas"
- line 227: "the CAZy"
- line 264: "Id 42.88"
- line 329: "are not engaged to form oligomeric interfaces in none of the GH31 members"
- line 340: "non-reducing"
- lines 406 and 423: "rely" should be "relies"
- lines 416 and 445: "added by" is used incorrectly.
- line 447: "being XacEgl9 the most active"
- line 456: "infect" should be "infects" (genus is singular)
- line 507: "Similarly, to that" (no comma needed)
- line 518: "unveils at mechanistic level"

A: The text was reviewed and checked for grammar and typo errors, including those aforementioned.

2. Line 166-174: I find this section logically a bit confusing: Isn't a lack of aromatic residues in the +3 and +5 subsites exactly what one would expect for an endo-dissociative enzyme, which XccXeg74 is? I don't think there is any need to invoke "molecular strategies complementary to the conventional stacking mechanism for carbohydrate binding". Certainly, the residues that are there may hydrogen-bond to the substrate, but this is - simply speaking - not enough to retain the substrate for endo-processivity. I would suggest replacing the sentence on lines 172-174

with something like "The lack of aromatic residues in subsites +3 and +5 of XccXeg74 is, therefore, consistent with its endo-dissociative mode of action on XyG."

A: Thank you for this observation. We changed the sentence as suggested.

3. Lines 189-237: As the novel CE is indicated to be a periplasmic enzyme, this section should come after the paragraph ending on line 250. In particular, the author show that the esterase is active on the primary hydrolysis products of the GH74 enzyme, so it can be inferred that the CE does not require any of the GHs, which are described in Line 251 and down, to act first.

A: We agree with the reviewer and placed the CE section after the paragraph ending on line 250.

4. Can the authors please confirm that they have been in contact with the curators of the CAZY Database and that a new CE family has, or will be, established?

A: We contacted the curators of the CAZY database and they recognized the esterase as the founding member of the new family CE20.

5. Lines 216 and 233: What is a "beta-Gal domain"? I don't see this described specifically, and it is unclear whether the authors mean that this is a GH, CBM, or another structural domain. Obviously, with a name like "beta-Gal", any predicted carbohydrate activity would be relevant to discuss in light of the XyG structure.

A: We initially named the domain as beta-gal due to its remote sequence similarity to ancillary domains from GH2 beta-galactosidases. However, the biological function of this domain remains elusive and, then, the use of beta-gal nomenclature could inappropriately indicate to an yet non-demonstrated function. Based on these arguments, it was renamed as X448 according to the suggestion from the CAZY database for this uncharacterized domain.

6. Line 207-209: This sentence is confusing, because the authors show explicitly that the esterase indeed has activity on oligosaccharides. Please revise to clarify the intention of this text.

A: We agree with the reviewer and removed the sentence.

7. Line 248-250: Please explicitly describe in a following sentence what was observed for the XAC1769 knock-out.

A: We included the required sentence: "The knockout of XAC1769 impaired the growth in the late log phase, indicating the importance of this transporter as the XyGOs concentration decreases in the medium (Supplementary Fig. 5)."

8. Line 350-352: This discussion of transcriptomic data comes before the major section on this (lines 373-376 and beyond). Perhaps some reference to "see section below" would be valuable to inform the reader that the glucosidase transcriptomic data is part of a larger dataset to follow.

A: Thank you for the insight. The reference was added.

9. Line 393 and/or line 469: Please include an explanation of the sources of these two types of XyG, with appropriate literature reference(s).

A: Examples of plants possessing such xyloglucan types and the respective references to them were added at both sentences.

10. Methods, "Protein crystallization, X-ray data collection and structure determination" section: Please use Privateer to validate all carbohydrate complexes and provide the output as a supplementary table (complementary to Supplementary Table 4). See and cite Nat Struct Mol Biol. 2015 Nov;22(11):833-4. doi: 10.1038/nsmb.3115

A: As requested, the Table was added in the Supplementary material (Supplementary Table 5). The summary of the analysis was also included in the crystallographic table (Supplementary Table 4). The literature reference was appropriately cited in the Supplementary material and in the Methods section on the main text.

End of comments.